# OmniFit: Bridging Modalities via Layer-Adaptive Token Compression for Omnimodal Large Language Models

Zining Wang [1 2]  Zhihang Yuan [3]  Yingjie Zhai [4]  Wenshuo Li [4]  Han Shu [4]  Ruihao Gong [1]  Jinyang Guo [1 5]
Xianglong Liu [1 2]

## Abstract

Emerging Omni-modal Large Language Models (OmniLLMs) enable real-time interaction across video, audio, and text but suffer from prohibitive computational costs due to the quadratic complexity of processing continuous streaming inputs. Existing token compression strategies remain suboptimal as they typically rely on biased modality-centric priors or enforce uniform retention policies, neglecting the heterogeneity across layers and the critical role of cross-modality alignment. To address these challenges, we propose OmniFit, a training-free framework that decouples interaction profiling from inference execution. OmniFit incorporates Layer-Adaptive Heterogeneity Profiling (LAHP) to dynamically allocate computational budgets based on layer-wise redundancy and modality preferences, preserving tokens according to the characteristics of each layer. Furthermore, we introduce Alignment-Rectified Token Selection (ARTS), a lightweight mechanism that efficiently identifies tokens semantically aligned with cross-modal cues. Extensive experiments on 3 model series across 10 benchmarks demonstrate that OmniFit establishes a new Pareto frontier, retaining **98%** of model performance with only **20%** token usage and achieves up to **2.31×** end-to-end inference speedup and **2.5×** VRAM saving, significantly outperforming state-of-the-art methods.

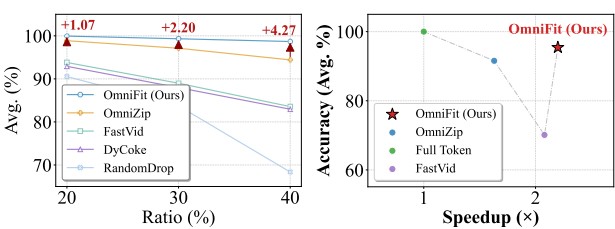

*Figure 1. (Left)* Average performance (%) *vs. retained token* ratios (%) on Qwen2.5-Omni-3B (Xu et al., 2025a) across different methods (Tao et al., 2025b; Shen et al., 2025; Tao et al., 2025a) on 10 benchmarks (as detailed in Sec. 6). *(Right)* Efficiency-performance trade-off on Qwen2.5-Omni. OmniFit (red star) establishes a new Pareto frontier, significantly outperforming baselines in both inference speed and accuracy.

## 1. Introduction

Recent advancements in Large Language Models (LLMs) (Yang et al., 2025a; Comanici et al., 2025) have evolved beyond text-centric paradigms, giving rise to Omni-modal Large Language Models (OmniLLMs) capable of natively processing streaming audio, visual, and textual content (Achiam et al., 2023; Xu et al., 2025a). Unlike traditional Multi-modal LLMs that handle static image-text pairs, OmniLLMs unify video, speech, and text into a shared Transformer backbone to enable end-to-end real-time interaction. However, this unified processing capability introduces substantial computational costs. The concatenation of high-resolution visual tokens and continuous audio frames results in significantly extended input sequences, exacerbating the quadratic complexity of the self-attention mechanism (Vaswani et al., 2017). This imposes prohibitive memory and computational burdens, restricting the deployment of OmniLLMs in real-time, resource-constrained scenarios.

Many methods (Xie et al., 2025; Huang et al., 2025; Li et al., 2026; Huang et al., 2024) have been proposed to alleviate this issue, with token compression emerging as a promising solution to mitigate these costs. While unimodal approaches (Bolya et al., 2022) and vision-language compression methods (Shen et al., 2025) have proven effective in their respective domains, they are fundamentally insufficient

---

[1]State Key Laboratory of Complex & Critical Software Environment [2]School of Computer Science and Engineering, Beihang University [3]Peking University [4]Huawei Technologies [5]School of Artificial Intelligence, Beihang University. Correspondence to: Xianglong Liu <xlliu@buaa.edu.cn>.

*Proceedings of the 43rd International Conference on Machine Learning*, Seoul, South Korea. PMLR 306, 2026. Copyright 2026 by the author(s).

for the omni-modal context, as they inherently overlook the dynamic synergy between audio and visual streams. Recent attempts to address this, such as OmniZip (Tao et al., 2025b) and EchoingPixels (Gong et al., 2025), explore audio-visual redundancy but face limitations. OmniZip adopts an audio-centric paradigm that compresses visual tokens based solely on audio signals, neglecting visual regions that are semantically critical but weakly aligned with the audio (*e.g.*, silent gestures). EchoingPixels (Gong et al., 2025), conversely, is a training-based method that relies on dense cross-modal attention for token selection. This approach operates on the premise that capturing semantic dependencies requires exhaustive global computation, introducing substantial overheads that often negate the latency benefits gained from token reduction.

To dismantle the limitations of static modality priors and dense attention computation, we first uncover two overlooked intrinsic properties of OmniLLMs: *(i) Layer-wise heterogeneity*. Challenging static assumptions, we observe distinct depth-dependent behaviors. Regarding redundancy, shallow layers necessitate dense information retention, whereas deep layers tolerate aggressive pruning. Furthermore, regarding modality preference, different layers exhibit fluctuating biases rather than uniform preference. *(ii) Cross-modal dominance*. Contradicting the necessity of dense attention, we uncover that cross-modal interactions are sparse and anchor-driven. Tokens with high cross-modal alignment scores often serve as essential semantic anchors, bridging modalities despite their low intra-modal significance. Crucially, since these patterns are inherent to the pre-trained backbone, they enable a lightweight, training-free framework that simply characterizes these behaviors, eliminating the need for costly retraining.

Driven by these insights, we introduce *OmniFit*, a training-free framework designed to profile these intrinsic properties without heavy computational overhead. Specifically, we propose *Layer-Adaptive Heterogeneity Profiling (LAHP)*, which translates the observed depth-dependent redundancy and modality biases into precise, layer-specific retention budgets. Complementing this, we introduce the *Alignment-Rectified Token Selection (ARTS)* mechanism. By leveraging a lightweight proxy metric to measure the projection alignment between tokens and global semantic anchors, ARTS effectively prioritizes cross-modal bridges. Notably, this design reduces the selection complexity from quadratic to linear $\mathcal{O}(ND)$, achieving up to a $42\times$ speedup in calculation overhead.

Extensive experiments on 3 OmniLLM families across 10 benchmarks demonstrate that OmniFit establishes a new Pareto frontier. As shown in Fig. 1, OmniFit consistently surpasses state-of-the-art (SOTA) methods, achieving a **4.27%** performance enhancement over baselines while retaining

$\geq 97\%$ of the original model's accuracy. Furthermore, OmniFit yields substantial efficiency gains, delivering up to a **2.31×** speedup in prefilling and a **1.39×** speedup in decoding for Qwen3-Omni-30B-A3B-Instruct (Yang et al., 2025a).

## 2. Related Works

**Omni-modal large language models.**

Recent advancements have shifted focus from text-centric models to Omni-modal LLMs (OmniLLMs) (Ye et al., 2025; Li et al., 2025; AI et al., 2025), enabling native end-to-end processing of interleaved text, audio, and vision (Li et al., 2024b; Hong et al., 2025; Fu et al., 2025). Representative architectures like GPT-4o (Achiam et al., 2023) and the Qwen-Omni series (Xu et al., 2025a;b) employ a unified Transformer to tokenize continuous video and speech streams for seamless real-time interaction. However, this inclusion of temporal modalities dramatically inflates sequence length, imposing severe memory and latency burdens due to the quadratic cost of attention. This bottleneck necessitates efficient token compression strategies tailored for omni-modal contexts.

**Token compression strategy.** Existing multi-modal compression typically relies on similarity-based (Tao et al., 2025a; Shen et al., 2025) or attention-based (He et al., 2024; Xing et al., 2024) pruning. However, these static vision-language methods often fail in dynamic omni-modal scenarios. Although recent works like OmniZip (Tao et al., 2025b) and EchoingPixels (Gong et al., 2025) explore audio-visual redundancy, OmniZip's audio-centric approach risks discarding text-relevant visual cues, while the training-based EchoingPixels incurs high overhead from dense cross-modal attention. In contrast, our training-free method introduces a lightweight alignment-rectified strategy that holistically evaluates token importance across all modalities, adaptively balancing layer-wise redundancy and modality preferences.

## 3. Preliminaries

**Architecture of OmniLLMs.** A typical OmniLLM (Tao et al., 2025b; Li et al., 2025; AI et al., 2025) consists of four primary components: a visual encoder, an audio encoder, a projector, and an LLM backbone. Functionally, the encoders and projector first convert raw inputs into text-aligned embeddings. These tokens are then organized into fixed-length time windows, where co-temporal audio-video features are concatenated chronologically to form a unified sequence. Finally, this sequence is processed by the LLM backbone, comprising a stack of Transformer layers (Vaswani et al., 2017) with self-attention and feed-forward networks (FFNs), to generate the response.

**DPC-KNN.** Widely adopted for token compression (Shao et al., 2025; Dhouib et al., 2025; Jin et al., 2024), DPC-KNN identifies cluster centers using two scalar metrics for each token $\mathbf{h}_i$. First, local density $\rho_i$ measures neighborhood concentration via $K$-nearest neighbors:

$$\rho_i = \exp\left(-\frac{1}{K}\sum_{j \in \text{KNN}(i)} \|\mathbf{h}_i - \mathbf{h}_j\|_2\right). \quad (1)$$

Second, the minimum distance $\delta_i$ quantifies the separation from denser regions:

$$\delta_i = \min_{j:\rho_j > \rho_i} \|\mathbf{h}_i - \mathbf{h}_j\|_2. \quad (2)$$

Tokens with high $\rho_i$ and $\delta_i$ are selected as centroids, while others are assigned to the nearest neighbor with higher density.

## 4. Motivation

Most existing token compression methods predominantly focus on video-only or audio-only contexts. However, such unimodal-centric approaches are insufficient for audio-video Omni-models, as they cannot leverage audiovisual cross-modal synergy, inevitably leading to modality misalignment. The few attempts to extend these methods to the Omni-domain (*e.g.*, OmniZip (Tao et al., 2025b) and EchoingPixels (Gong et al., 2025)) typically resort to direct adaptations, treating the model as a structurally uniform stack. In this study, we have identified two critical factors overlooked by these approaches: layer-wise heterogeneity (Sec. 4.1) and cross-modal dominance (Sec. 4.2). Modeling these factors is critical, as they determine the effective boundary between redundancy reduction and semantic preservation.

### 4.1. Overlooked Layer-Wise Heterogeneity

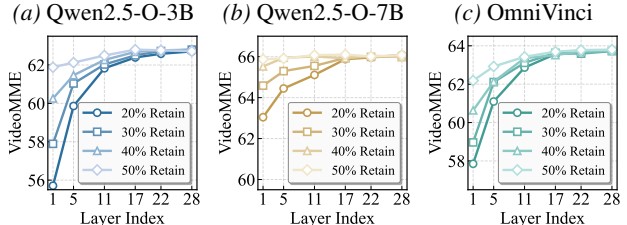

*(a)* Qwen2.5-O-3B    *(b)* Qwen2.5-O-7B    *(c)* OmniVinci

*Figure 2.* Performance across Qwen2.5-Omni (Xu et al., 2025a) (*i.e.*, (a)-(b)) and OmniVinci (Ye et al., 2025) (*i.e.*, (c)) on VideMME (Fu et al., 2025) benchamrk under different retained token ratios (*i.e.*, 20% to 50%).

*Observation (i): Depth-dependent token redundancy.* Recent token compression strategies (Tao et al., 2025b; Gong et al., 2025) typically treat the transformer backbone as a topologically homogeneous stack. However, existing studies (Xing et al., 2024; He et al., 2024; Zhuang et al., 2025) reveal that token redundancy varies significantly with depth.

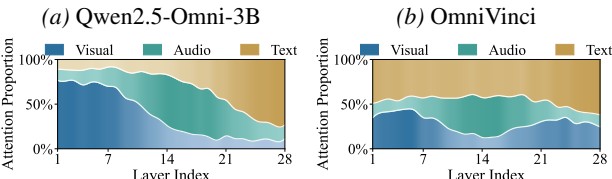

*(a)* Qwen2.5-Omni-3B    *(b)* OmniVinci

*Figure 3.* Visualization of layer-wise modality attention proportions. We compare the averaged attention distribution across layers for (a) Qwen2.5-Omni-3B (Xu et al., 2025a) and (b) OmniVinci (Ye et al., 2025), calculated over 1,024 varied data samples. Color intensity encodes magnitude.

Mirroring this, we empirically verify that depth-dependent redundancy is strictly preserved in OmniLLMs. Specifically, we independently prune tokens at a target layer with varying retention ratios to evaluate the resulting performance impact. As visualized in Fig. 2, a consistent trend emerges across all models: When reducing the ratio of retained tokens, early layers incur significantly more severe performance drops than deep layers. We attribute this to two factors. First, shallow layers capture global dependencies where most tokens are essential, whereas deep layers focus on sparse key tokens (Rao et al., 2021). Second, information loss in shallow layers leads to cumulative error propagation, while deep pruning limits this impact (Bolya et al., 2022). Consequently, layer-independent strategies risk over-compressing sensitive shallow layers while under-compressing redundant deep layers.

*Insight (i):* An optimal compression strategy must be depth-adaptive: conservative in sensitive shallow layers and aggressive in redundant deep layers.

*Observation (ii): Layer-wise modality preference.* Existing approaches exhibit a dichotomy in handling multi-modal tokens: some treat all modalities as equally important (*i.e.*, uniform treatment (Gong et al., 2025)), while others assume a static dominance of specific modalities (*e.g.*, audio-centric (Tao et al., 2025b)). We identify that both perspectives are oversimplified. As visualized in Fig. 3, our analysis reveals that attention preference is neither uniform nor static, but exhibits distinct layer-wise heterogeneity dependent on the model architecture. This stems from the network's hierarchical transition from sensory alignment to abstract reasoning. As a result, rigid compression strategies that ignore these dynamics are fundamentally suboptimal, as they cannot match the fluctuating information density of different modalities.

*Insight (ii):* Token compression strategies should be modality-adaptive by aligning with layer-wise attention preferences, *e.g.*, retaining more visual tokens in layers exhibiting high visual attention.

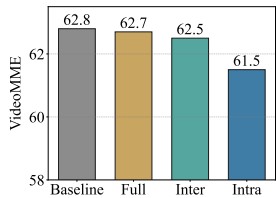

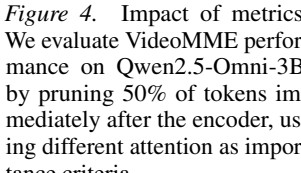

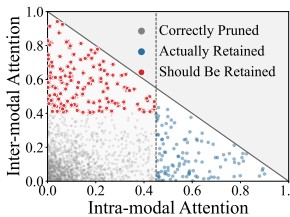

*Figure 4.* Impact of metrics. We evaluate VideoMME performance on Qwen2.5-Omni-3B by pruning 50% of tokens immediately after the encoder, using different attention as importance criteria.

*Figure 5.* Visualization of the Metric Mismatch. We aggregate 1,500 samples and plot the distribution of token importance on Qwen2.5-Omni-3B. The vertical line denotes the pruning threshold.

## 4.2. Invisible yet Critical Cross-Modal Dominance

*Observation (iii): Cross-modality matters.* Ideally, token compression should be guided by *Ground-Truth Attention*, but IO-aware optimizations like FlashAttention (Dao et al., 2022; Dao, 2023), which are widely equipped among mainstream OmniLLMs, render these weights invisible during inference. Consequently, many existing methods (Tao et al., 2025b; Jin et al., 2024; Shao et al., 2025) resort to *Intra-modal Estimation* as a proxy (*e.g.*, distance between two tokens from the same modality in Eq.( 2)), yet our experiments in Fig. 4 reveal a critical disparity: Inter-modal interaction characterizes token importance significantly better than intra-modal priors. Intra-modal guidance causes catastrophic collapse, confirming that inter-modal connection is the true determinant of semantic value in OmniLLMs (Chefer et al., 2021). Fig. 5 visualizes the consequence of ignoring this dominance: relying on intra-modal metrics leads to the erroneous pruning of meaningful tokens. As shown, invisible anchor tokens (*i.e.*, red points), which are crucial for inter-modal alignment, are systematically discarded simply because they lack intra-modal saliency.

*Insight (iii):* To preserve reasoning capabilities and semantic alignment without ground-truth attention, metric design should prioritize cross-modal saliency.

## 5. OmniFit

Based on the above observations and insights, we propose *OmniFit*, an *efficient training-free* framework composed of two key components, as illustrated in Fig. 6: (i) *layer-adaptive heterogeneity profiling (LAHP)* (Sec. 5.1), which profiles layer-wise token redundancy and modality preferences to serve as a strategic guidance for inference; and (ii) *alignment-rectified token selection (ARTS)* (Sec. 5.2), a mechanism that incorporates cross-modal interaction priors to preserve semantically aligned tokens and critical information. The seamless integration of these two components ensures both effectiveness and efficiency.

## 5.1. Layer-Adaptive Heterogeneity Profiling

In light of *Insight (i)* and *Insight (ii)* in Sec. 4.1, we present a *layer-adaptive heterogeneity profiling (LAHP)* method, which is composed of *Token Redundancy Profiling* and *Modality Preference Profiling*. The former quantifies the token redundancy of each layer to implement a progressive compression strategy based on layer-wise information density. The latter dynamically allocates appropriate retention rates to different modalities according to the unique attention preference of each layer, thereby realizing a holistic layer-adaptive token compression strategy.

**Token Redundancy Profiling.** To address the *depth-dependent token redundancy* in *Observation (i)*, we quantify the information density of the $l$-th Transformer block using its output states $\mathbf{X}^{(l)} \in \mathbb{R}^{N \times d}$, where $N$ is the sequence length and $d$ is the hidden dimension. Specifically, We perform Singular Value Decomposition (SVD) on the $\mathbf{X}^{(l)}$ to obtain the singular values $\{\sigma_i\}$. We explicitly calculate the *Effective Rank* $k_{\text{eff}}^{(l)}$ as the minimum number of dimensions required to preserve a cumulative energy ratio $\delta$ (e.g., 0.9):

$$k_{\text{eff}}^{(l)} = \min \left\{ k \ \middle| \ \frac{\sum_{i=1}^{k} \sigma_i^2}{\sum_{j=1}^{d} \sigma_j^2} > \delta \right\}. \tag{3}$$

A lower $k_{\text{eff}}^{(l)}$ indicates feature space collapse, signifying high redundancy suitable for aggressive pruning[1].

Guided by the layer-wise redundancy, we formulate the final layer-wise retention rate $r^{(l)}$ for a model with $L$ layers:

$$r^{(l)} = \xi \cdot \left( \mu \cdot \frac{\Psi^{(l)}}{\frac{1}{L} \sum_{j=1}^{L} \Psi^{(j)}} \right), \quad \text{where } \Psi^{(l)} = \prod_{i=1}^{l} \frac{k_{\text{eff}}^{(i)}}{d}. \tag{4}$$

Here, $\mu$ is the global target compression ratio (e.g., 0.5), and $\Psi^{(l)}$ ensures a monotonic decay (*i.e.*, progressive compression) shape based on cumulative effective ranks.

To offset the quadratic overhead inherent to non-uniform pruning[2] and guarantee strict computational efficiency, we introduce and solve for the penalty $\xi$ under the constraint $\sum \mathcal{C}(r^{(l)} N) \leq \mathcal{C}_{\text{Uniform}}$. Modeling the layer cost as $\mathcal{C}(n) = c_1 n + c_2 n^2$ ($c_1$: linear, $c_2$: quadratic coeffs), we derive the closed-form solution:

$$\xi = \frac{-\mathcal{B} + \sqrt{\mathcal{B}^2 + 4\mathcal{A} \cdot \mathcal{C}_{\text{Uniform}}}}{2\mathcal{A}}. \tag{5}$$

where $\mathcal{A} = \sum l c_2 (\hat{r}^{(l)} N)^2$ and $\mathcal{B} = \sum_l c_1 (\hat{r}^{(l)} N)$ aggregate the quadratic and linear costs of the unscaled profile

---

[1]The Eckart-Young-Mirsky theorem bounds error by discarded singular values; rapid spectral decay guarantees negligible information loss. Proof in Appendix A.

[2]Mathematically, due to convexity of attention ($n^2$), Cauchy-Schwarz implies variance strictly increases FLOPs. Proof in Appendix A.

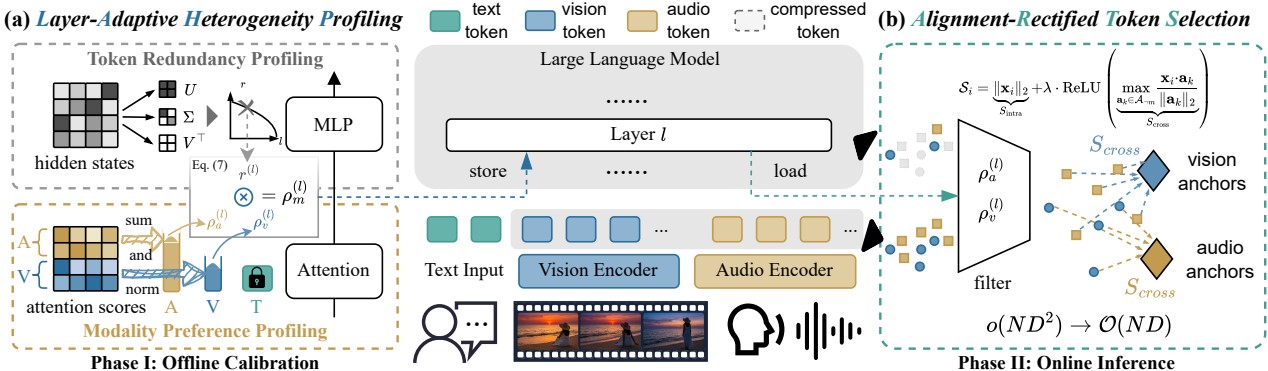

Figure 6. Overview of *OmniFit*. At inference, use a text token (*i.e.*, ■ above) at the $l$-th LLM layer as an example. *(a)* We calibrate the vision retention rate $r_v^{(l)}$ by combining the $r^l$ and $\rho_v^{(l)}$ derived from *token redundancy profiling* and *modality preference profiling*, respectively. *(b)* During inference, we calculate the importance scores $\mathcal{S}_i$ for each vision token. We then load $r_v^{(l)}$ and compress the video tokens according to the scores.

$\hat{r}^{(l)}$ (*i.e.*, Eq. (4) without penalty factor $\xi$). Crucially, since these terms rely solely on calibration statistics, $\xi$ is computed directly during the calibration phase with negligible overhead. This solution guarantees $\xi < 1$, maximizing information density without exceeding the baseline budget.

**Modality Preference Profiling.** Motivated by *Observation (ii)*, we define the *Modality Preference Score* $\rho_m^{(l)}$ as the average attention density received by modality $m$. It aggregates the attention mass from the entire context, normalized by token count $N_m$, $m \in [a, v, t]$:

$$\rho_m^{(l)} = \frac{1}{N_m} \sum_{j \in m} \underbrace{\left( \sum_{i=1}^{N} \bar{\mathbf{A}}_{i,j}^{(l)} \right)}_{\text{Attention on Token } j} . \qquad (6)$$

Here, $\bar{\mathbf{A}}_{i,j}^{(l)}$ is the average attention probability after `Softmax`. Normalization by $N_m$ ensures the metric captures intrinsic information density, eliminating bias towards longer sequences (*e.g.*, vision).

Then we employ a parameter-free reallocation strategy. We first derive the layer's total token budget $K^{(l)} = r^{(l)}N$ using the global rate $r^{(l)}$ from Eq. (4). Given the high semantic density and low redundancy of text tokens, we preserve them in full ($r_t = 1$). The residual budget $K_{\text{res}} = K^{(l)} - N_t$ is then distributed between visual and audio tokens proportional to their densities:

$$r_m^{(l)} = \rho_m^{(l)} \cdot \frac{K_{\text{res}}}{\rho_v^{(l)} N_v + \rho_a^{(l)} N_a}, \quad \forall m \in v, a. \qquad (7)$$

This linear scaling ensures that modalities with higher attention density ($\rho_m^{(l)}$) are assigned a higher retention rate ($r_m^{(l)}$), thereby protecting critical information while aggressively pruning less relevant modalities. In this way, LAHP provides a theoretically grounded, layer-specific allocation.

### 5.2. Alignment-Rectified Token Selection

To efficiently capture tokens critical for cross-modal alignment without incurring $O(N^2)$ overhead (*Observation (iii)*), we propose *Alignment-Rectified Token Selection (ARTS)*. Specifically, immediately following the encoder, we utilize DPC-KNN (Sec. 3) to distill the output tokens into a compact set of global anchors $\mathcal{A}_m$ for each modality. These pre-computed anchors are stored to serve as lightweight, layer-agnostic references for cross-modal relevance. During the forward pass, we calculate the importance score $\mathcal{S}_i$ for a token $\mathbf{x}_i \in m$ by fusing its intrinsic magnitude with its alignment to the opposing modality $\neg m$ (represented by anchors $\mathcal{A}_{\neg m}$):

$$\mathcal{S}_i = \underbrace{\|\mathbf{x}_i\|_2}_{S_{\text{intra}}} + \lambda \cdot \text{ReLU}\left( \underbrace{\max_{\mathbf{a}_k \in \mathcal{A}_{\neg m}} \frac{\mathbf{x}_i \cdot \mathbf{a}_k}{\|\mathbf{a}_k\|_2}}_{S_{\text{cross}}} \right) \qquad (8)$$

The first term, $S_{\text{intra}}$, captures structural significance via the $L_2$ norm. The second term, $S_{\text{cross}}$, measures the maximum cosine similarity to the opposing anchors. Crucially, this term is dynamically weighted by the Modality Preference Score of the opposing modality (*i.e.*, $\rho_{\neg m}^{(l)}$). This ensures that when the layer heavily attends to the opposing modality, tokens aligned with it are prioritized. We retain the top-$k$ tokens based on $\mathcal{S}_i$ and merge the rest (Sec. I).

## 6. Experiments

### 6.1. Setups

**Models and datasets.** We choose 3 series of OmniLLMs to evaluate *OmniFit*: Qwen2.5-Omni (Xu et al., 2025a), OmniVinci (Ye et al., 2025), and Qwen3-Omni-30B-A3B-Instruct (Xu et al., 2025b). For audio-video tasks, we use

*Table 1.* Performance comparisons for Qwen2.5-Omni-3B (Xu et al., 2025a) across various token retained ratios. We mark the practical token retained ratio (*i.e.*, "*Retain x% Rokens*") in the table. For each method, we compute the score proportion relative to the full tokens (*i.e.*, original model without token compression) across benchmarks, and then compute the average value in the "(Avg. (%))" column. The best and second-best results are highlighted in **bold** and underlined formats, respectively.

| Method | Audio-Video Tasks | | | | | | | Video-Only Tasks | | | Avg. (%) |
|---|---|---|---|---|---|---|---|---|---|---|---|
| | AVUTBench | VideoMME | WorldSense | Daily-Omni | Video-Holmes | UNO-Bench | OmniBench | MVBench | EgoSchema | MLVU | |
| Full Tokens | 62.5 | 62.8 | 46.0 | 59.8 | 25.8 | 26.5 | 52.2 | 68.7 | 61.4 | 68.9 | 100.00 |
| *Retain 40% Tokens* | | | | | | | | | | | |
| RandomDrop | 57.1 | 60.9 | 40.2 | 52.1 | 18.8 | 22.1 | 49.5 | **67.5** | 60.5 | 65.2 | 90.54 |
| FastVid (Shen et al., 2025) | 58.9 | 61.5 | 41.6 | 55.9 | 20.1 | 23.5 | 50.1 | **68.5** | 61.4 | 68.7 | 93.81 |
| DyCoke (Tao et al., 2025a) | 58.1 | 61.2 | 42.8 | 55.1 | 20.2 | 22.9 | 50.0 | 67.9 | 61.0 | 65.4 | 92.92 |
| OmniZip (Tao et al., 2025b) | 60.8 | 62.7 | 44.8 | 59.5 | **25.8** | 26.1 | **52.0** | - | - | - | 98.87 |
| OmniFit (*Ours*) | **62.3** | **62.9** | **45.7** | **60.1** | **25.8** | **26.6** | 51.9 | **68.5** | **61.5** | **68.9** | **99.94** |
| *Retain 30% Tokens* | | | | | | | | | | | |
| RandomDrop | 53.2 | 59.6 | 35.3 | 50.3 | 17.6 | 16.5 | 46.9 | 63.8 | 56.5 | 62.8 | 83.72 |
| FastVid (Shen et al., 2025) | 56.8 | 61.2 | 38.9 | 51.8 | 20.1 | 18.4 | 48.1 | 66.2 | **61.4** | 65.1 | 88.99 |
| DyCoke (Tao et al., 2025a) | 55.4 | 60.5 | 39.8 | 51.8 | 19.6 | 18.5 | 47.8 | 64.1 | 59.2 | 65.1 | 87.97 |
| OmniZip (Tao et al., 2025b) | 59.5 | **62.5** | 42.7 | 58.6 | 25.3 | 25.9 | 51.4 | - | - | - | 97.12 |
| OmniFit (*Ours*) | **61.9** | **62.5** | **45.4** | **60.0** | **25.6** | **26.4** | **51.5** | **68.3** | **61.4** | **68.0** | **99.32** |
| *Retain 20% Tokens* | | | | | | | | | | | |
| RandomDrop | 48.5 | 50.1 | 20.9 | 40.9 | 10.8 | 11.2 | 40.3 | 59.4 | 51.5 | 55.7 | 68.37 |
| FastVid (Shen et al., 2025) | 53.9 | 60.1 | 35.2 | 45.3 | 19.9 | 12.0 | 47.5 | 65.4 | 61.0 | 64.4 | 83.56 |
| DyCoke (Tao et al., 2025a) | 51.6 | 58.8 | 35.9 | 44.5 | 19.1 | 16.1 | 45.6 | 61.2 | 58.6 | 64.9 | 82.95 |
| OmniZip (Tao et al., 2025b) | 57.2 | 60.5 | 41.6 | 57.5 | 24.6 | 25.0 | 50.5 | - | - | - | 94.41 |
| EchoingPixels (Gong et al., 2025) | - | 60.7 | 45.0 | **60.6** | - | - | - | - | - | **68.3** | **98.74** |
| OmniFit (*Ours*) | **61.5** | **62.0** | **45.1** | 59.8 | **25.5** | **26.1** | **51.2** | **68.0** | **61.2** | 67.2 | 98.68 |

7 benchmarks for evaluation: AVUTBench (Yang et al., 2025b), VideoMME (Fu et al., 2025), WorldSense (Hong et al., 2025), Daily-Omni (Zhou et al., 2025b), Video-Holmes (Cheng et al., 2025), UNO-Bench (Chen et al., 2025), and OmniBench (Li et al., 2024b). We adopt 3 benchmarks for video-only tasks: MVBench (Li et al., 2024a), EgoSchema (Mangalam et al., 2023), and MLVU (Zhou et al., 2025a). lmms-eval (Zhang et al., 2024) is utilized to perform the majority of the above evaluation.

**Baselines.** We benchmark OmniFit against the state-of-the-art omni-modal method OmniZip (Tao et al., 2025b), alongside EchoingPixels (Gong et al., 2025) (reporting original results due to code unavailability). Furthermore, we extend unimodal token compression strategies, DyCoKe (Tao et al., 2025a) and FastVid (Shen et al., 2025), to the omni-modal framework for a rigorous comparison, and include Random-Drop (*i.e.*, randomly drop tokens after the encoder according to the retained token ratio) as a naive lower bound to assess the necessity of sophisticated selection mechanisms.

**Implementation.** We employ 1024 samples randomly picked from AVQA (Yang et al., 2022) and Ola (Liu et al., 2025) datasets to calibrate $r_m^{(l)}$ (Eq. (7)). We set $\lambda = 1.5$ to prioritize the cross-modality. More results are in Appendix.

### 6.2. Evaluation

**Comparison with baselines**. We benchmark OmniFit against baselines on Qwen2.5-Omni-3B (Xu et al., 2025a). As shown in Tab. 1, prior methods, such as FastVid (Shen et al., 2025), DyCoke (Tao et al., 2025a), and OmniZip (Tao et al., 2025b), struggle to balance performance and effi-ciency, especially at low *token retained* ratio ($\leq 30\%$). Specifically, when extending uni-modal methods (*i.e.*, FastVid and DyCoke) to the omni-modal setting, they suffer a severe accuracy loss of over 16% on average at a 20% retention ratio. This significant degradation indicates that independently compressing distinct modalities neglects cross-modal dependencies, leading to modality mismatch and information loss. Similarly, the omni-modal baseline OmniZip also exhibits limitations, with a performance drop of over 4% at the same ratio. This suggests that treating all layers uniformly as audio-dominant is insufficient for capturing complex, layer-specific modality preferences. In contrast, our method simultaneously accounts for token redundancy and layer-wise modality preferences. Even when aggressively compressing to retain only 20% of tokens, OmniFit maintains 98.68% of the original model's performance. Furthermore, at higher retention ratios, our method consistently outperforms OmniZip, surpassing it by 1.07% at the 40% ratio and 2.20% at the 30% ratio. Notably, at 20% sparsity, OmniFit even achieves performance comparable to EchoingPixels (Gong et al., 2025), a training-based method, despite being training-free. These findings validate the superiority of our method across different skipping ratios compared with existing SOTA approaches.

**Evaluation under low retained token ratios.** We further verified the robustness of OmniFit under extreme compression regimes ($\mu \leq 10\%$). As shown in Tab. 2, our method consistently outperforms SOTA. Remarkably, at a stringent 5% token ratio, OmniFit even surpasses the training-based method EchoingPixels (Gong et al., 2025). For instance, on VideoMME, WorldSense, and Daily-Omni, OmniFit sur-

*Table 2.* Performance comparison under extreme compression regimes ($\leq 10\%$). All methods are evaluated on Qwen2.5-Omni-3B. The best results are highlighted in bold.

| Method | VideoMME | WoorldSense | Daily-Omni | MLVU |
|---|---|---|---|---|
| Full Tokens | 62.8 | 46.0 | 59.8 | 68.9 |
| *Retain 10% Tokens ($\mu = 0.1$)* | | | | |
| OmniZip (Tao et al., 2025b) | 58.1 | 40.5 | 56.3 | - |
| EchoingPixels (Gong et al., 2025) | 58.4 | **43.5** | 57.5 | **67.4** |
| OmniFit (*Ours*) | **58.8** | 43.4 | **58.3** | 66.3 |
| *Retain 5% Tokens ($\mu = 0.05$)* | | | | |
| OmniZip (Tao et al., 2025b) | 53.9 | 36.5 | 49.3 | - |
| EchoingPixels (Gong et al., 2025) | 55.7 | 40.9 | 52.8 | **66.1** |
| OmniFit (*Ours*) | **55.9** | **41.1** | **54.1** | 65.7 |

passes EchoingPixels with scores of 55.9 *vs.* 55.7, 41.1 *vs.* 40.9, and 54.1 *vs.* 52.8, respectively. This is attributed to our *layer-adaptive heterogeneity profiling (LAHP)*, which handles token redundancy layer-by-layer rather than aggressively discarding information at the input, thereby preserving essential semantic cues even under tight token budgets.

**Comparison across models.** In Tab. 3, we evaluate our method across multiple models. On Qwen2.5-Omni-7B, our approach retains 97.28% of the original performance at an aggressive token retained ratio of 20%, proving its superior scaling capability. This effectiveness generalizes to varying architecture designs, as evidenced by a 95.87% retention on OmniVinci, which adopts different backbones and training strategies from Qwen2.5-Omni. Furthermore, on the Qwen3-Omni-30B-A3B-Instruct, OmniFit sustains 93.46% retention (vs. OmniZip's 90.28%), proving its effectiveness even on complex Mixture-of-Experts (MoE) architectures. Taken together, these results highlight the effectiveness and universality of our method in identifying redundant tokens of different modalities and across different layers.

### 6.3. Efficiency Discussion

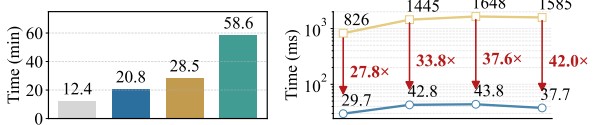

*Figure 7.* *(Left)* $r_m$ (Eq. (7)) calibration time. *(Right)* calculation time of attention score (yellow) *vs.* all token's $\mathcal{S}_i$ (Eq. (8)) (blue). From left to right are Qwen2.5-Omni-3B, Qwen2.5-Omni-7B, OmniVinci, and Qwen3-Omni-30B-A3B-Instruct.

**Calibration efficiency.** As illustrated in Fig. 7 *(left)*, we evaluate the calibration time of OmniFit for different models on 8×H800 GPUs. We can observe that OmniFit can calibrate OmniLLMs in an hour, even in MoE models like Qwen3-Omni-30B-A3B-Instruct. For dense models like Qwen2.5-Omni and OmniVinci, the calibration process can be done in half an hour. This demonstrates the efficiency of our calibration method, ensuring its practical application.

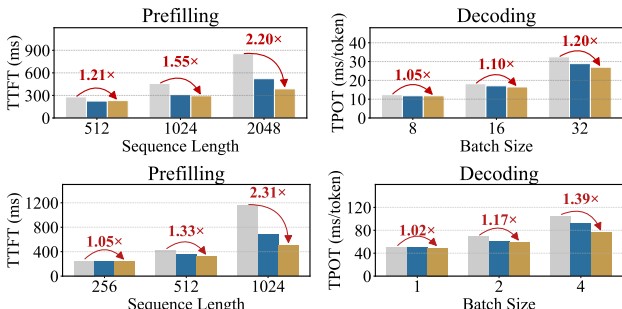

*Figure 8.* Inference speed of origin model (gray), OmniZip (blue), and ours (yellow) for *(Upper)* Qwen2.5-Omni-7B (Xu et al., 2025a) and *(Lower)* Qwen3-Omni-30B-A3B-Instruct (Xu et al., 2025b) on a single H800 GPU. The batch size for prefilling is 8, and the sequence length for decoding is 1024. According to Tab. 1 and Tab. 3, we set the retained token ratio to 30% for OmniZip and 10% for ours, respectively, to ensure comparable accuracy levels.

**Inference efficiency.** Next, we investigate the practical inference speedup. We first compared the latency of the online Alignment-Rectified Token Selection process against the standard full attention score calculation across various model scales. As shown in Fig. 7 *(right)*, our method demonstrates negligible overhead, achieving significant speedups ranging from 27.8× for Qwen2.5-Omni-3B to 42.0× for the deep Qwen3-Omni-30B-A3B model. This substantial performance gap confirms that the cost of our token importance evaluation is minimal compared to the quadratic complexity of dense attention, ensuring the theoretical benefits of real-world inference acceleration. We then benchmark the end-to-end real speedup. As shown in Fig. 8, OmniFit attains a 2.20× 2.31× prefilling speedup on Qwen2.5-Omni-7B and Qwen3-Omni-30B-A3B-Instruct, respectively. In the decoding phase, it still delivers up to 1.20× and 1.39× real acceleration. These results demonstrate that our method outperforms SOTA methods in both accuracy (Tab. 1 and Tab. 3) and efficiency. Notably, we observe that the acceleration gains amplify as the sequence length and batch size increase. This scalability stems from OmniFit's dual-optimization capability: it not only lowers redundant calculations but also reduces GPU memory usage, thereby simultaneously alleviating bottlenecks in both computation-bound (long-context prefilling) and memory-bound (high-throughput decoding).

**Memory footprint.** As shown in Tab. 4, OmniFit reduces the 7B model's memory usage by 2.5×. Crucially, for the 30B model, it prevents OOM, successfully fitting a single H800-80GB GPU on WoorldSense.

*Table 4.* Real-world VRAM.

| Model | Full | OmniFit |
|---|---|---|
| Qwen-7B | 35.7G | **14.5G** (2.5×) |
| Qwen-30B | **OOM** | **70.2G** (Feasible) |

### 6.4. Ablation Studies

**Effect of each component.** We evaluate each component of OmniFit and use RandomDrop as our baseline. As shown in

*Table 3.* Performance comparisons across different models. Operating with a 20% retention ratio ($\mu = 0.2$), OmniFit proves its robustness across three dimensions: scaling (Qwen2.5-Omni-7B), diverse architectures (OmniVinci), and MoE mechanisms (Qwen3-Omni-30B-A3B-Instruct). Results show consistent superiority over existing baselines.

| Method | Audio-Video Tasks | | | | | Video-Only Tasks | | | Avg. (%) |
|---|---|---|---|---|---|---|---|---|---|
| | VideoMME | WorldSense | Daily-Omni | Video-Holmes | OmniBench | MVBench | EgoSchema | MLVU | |
| *Qwen2.5-Omni-7B (Xu et al., 2025a)* | | | | | | | | | |
| Full Tokens | 66.1 | 47.5 | 62.3 | 27.1 | 56.1 | 70.3 | 68.6 | 68.5 | 100.00 |
| RandomDrop | 52.3 | 30.3 | 40.8 | 17.6 | 45.0 | 59.8 | 60.2 | 53.2 | 75.51 |
| FastVid (Shen et al., 2025) | 55.7 | 33.4 | 42.6 | 20.9 | 48.1 | 64.3 | 65.9 | 62.1 | 83.00 |
| OmniZip (Tao et al., 2025b) | 64.4 | 44.2 | 59.9 | 23.2 | 53.9 | - | - | - | 93.66 |
| OmniFit *(Ours)* | **65.1** | **46.6** | **61.8** | **25.9** | **55.5** | **68.9** | **65.9** | **64.3** | **97.28** |
| *OmniVinci (Ye et al., 2025)* | | | | | | | | | |
| Full Tokens | 71.3 | 48.3 | 66.0 | 31.2 | 61.5 | 71.1 | 69.2 | 68.4 | 100.00 |
| RandomDrop | 65.2 | 29.9 | 48.3 | 20.5 | 47.7 | 60.9 | 61.2 | 51.7 | 77.43 |
| FastVid (Shen et al., 2025) | 67.5 | 35.5 | 51.0 | 23.3 | 51.1 | 64.5 | 68.0 | 60.2 | 85.03 |
| OmniZip (Tao et al., 2025b) | 70.1 | 44.2 | 63.5 | 26.4 | 58.9 | - | - | - | 93.29 |
| OmniFit *(Ours)* | **70.8** | **46.2** | **64.9** | **28.9** | **60.4** | **65.1** | **68.2** | **63.4** | **95.87** |
| *Qwen3-Omni-30B-A3B-Instruct (Xu et al., 2025b)* | | | | | | | | | |
| Full Tokens | 73.5 | 54.0 | 72.9 | 56.4 | 67.5 | 74.3 | 72.4 | 75.2 | 100.00 |
| RandomDrop | 62.7 | 36.3 | 51.5 | 39.2 | 59.8 | 65.4 | 63.8 | 62.5 | 80.07 |
| FastVid (Shen et al., 2025) | 65.1 | 40.2 | 58.8 | 45.4 | 62.2 | 68.9 | **66.1** | 65.0 | 85.85 |
| OmniZip (Tao et al., 2025b) | 67.8 | 46.2 | 66.3 | 50.3 | 63.1 | - | - | - | 90.28 |
| OmniFit *(Ours)* | **70.1** | **51.2** | **70.6** | **52.1** | **65.0** | **70.1** | **65.9** | **65.1** | **93.46** |

*Table 5.* Ablation results for each component of OmniFit on Qwen2.5-Omni-3B. The combination of LAHP and ARTS (Ours) yields optimal performance across all benchmarks on Qwen2.5-Omni-3B, highlighting the necessity of combining layer-adaptive budgeting with alignment-rectified selection, particularly at a strict 20% retention budget.

| Method | VideoMME | WorldSense | Daily-Omni | MVBench | MLVU |
|---|---|---|---|---|---|
| Full Tokens | 62.8 | 46.0 | 59.8 | 68.7 | 68.9 |
| *Retain 30% Tokens ($\mu = 0.3$)* | | | | | |
| RandomDrop | 59.6 | 35.3 | 50.3 | 63.8 | 62.8 |
| RandomDrop w/ LAHP | 62.0 | 42.0 | 56.6 | 67.5 | 65.6 |
| ARTS | 61.1 | 41.5 | 55.4 | 64.5 | 65.1 |
| ARTS w/ TRP | 61.8 | 42.5 | 56.5 | 67.9 | 67.5 |
| ARTS w/ LAHP *(Ours)* | 62.5 | 45.4 | 60.0 | 68.3 | 68.0 |
| *Retain 20% Tokens ($\mu = 0.2$)* | | | | | |
| RandomDrop | 50.1 | 20.9 | 40.9 | 59.4 | 55.7 |
| RandomDrop w/ LAHP | 55.3 | 30.1 | 51.5 | 65.5 | 63.9 |
| ARTS | 55.4 | 28.6 | 48.5 | 61.8 | 60.1 |
| ARTS w/ TRP | 58.9 | 36.5 | 55.4 | 67.7 | 67.0 |
| ARTS w/ LAHP *(Ours)* | 62.0 | 45.1 | 59.8 | 68.0 | 67.2 |

*Table 6.* Ablation results of using different calibration datasets. Profiling with 1024 samples from different sources yields consistently stable performance (e.g., less than 0.5 variance on Daily-Omni). This confirms OmniFit's dataset-agnostic stability and robust calibration process.

| Dataset | VideoHomles | Daily-Omni | MVBench | MLVU |
|---|---|---|---|---|
| Full Tokens | 25.8 | 59.8 | 68.7 | 68.9 |
| VideoMME | 25.6 | 60.3 | 68.3 | 68.1 |
| WorldSense | 25.8 | 59.8 | 68.3 | 67.6 |
| OmniBench | 25.4 | 60.0 | 68.4 | 68.1 |
| AVQA+Ola *(Ours)* | 25.6 | 60.0 | 68.3 | 68.0 |

Tab. 5, LAHP, which incorporates *token redundancy profiling* (TRP) and *modality preference profiling* (MPP), significantly improves both RandomDrop and ARTS. Specifically, TRP proves indispensable at lower retention rates (*e.g.*, 20%) by dynamically preserving budgets for information-dense layers, while MPP is critical for omni-modal tasks to balance audio-visual contributions based on semantic relevance. Furthermore, ARTS outperforms RandomDrop across all benchmarks. By combining ARTS and LAHP, OmniFit resolves the heterogeneity of each layer and filters important tokens precisely, achieving performance on par with the full-token baseline even under aggressive compression.

**Robustness to calibration data.** We also investigate the effect of different calibration datasets. In Tab. 6, we can observe that using different datasets for calibration yields consistently stable performance across all evaluation benchmarks, with negligible fluctuations (*e.g.*, less than 0.5 variance on Daily-Omni). These results indicate that OmniFit is robust and not sensitive to the choice of dataset.

**Robustness to Calibration Data Size** $N_{calib}$. We investigate the minimum data requirement for reliable layer-adaptive profiling. The results in Table 7 reveal a clear saturation point at $N_{calib} = 256$. While extremely small subsets (*e.g.*, 16 or 64) lead to suboptimal profiling, increasing the sample size beyond 256 yields negligible performance gains. This rapid convergence highlights the robustness of our approach, proving that reliable layer-adaptive profiling can be

achieved with a strictly limited calibration budget.

*Table 7.* Effect of Calibration Sample Size. The profiling converges rapidly, requiring minimal data.

| $N_{calib}$ | 16 | 64 | 256 | 1024 | 2048 |
|---|---|---|---|---|---|
| **VideoMME** | 58.2 | 60.5 | 61.9 | **62.0** | 62.0 |
| **Daily-Omni** | 55.4 | 58.1 | 59.7 | 59.8 | **59.9** |

**Effectiveness of the Cumulative Product Schedule.** Progressive token pruning strictly requires a monotonically decreasing retention schedule. By defining the retention profile via a cumulative product of normalized effective ranks ($\Psi^{(l)} = \prod_{i=1}^{l}(k_{eff}^{(i)}/d)$), LAHP natively guarantees this property. Unlike direct per-layer normalization, which may infeasibly assign larger budgets to deeper layers, our formulation explicitly avoids this risk. As shown in Table X, our data-driven schedule consistently outperforms heuristic monotonic baselines (Linear and Cosine Decay) under a 20% budget, confirming its superiority in capturing actual layer-wise redundancy.

*Table 8.* Comparison of Layer Schedules. Evaluated on Qwen2.5-Omni-3B at a 20% retention rate. Our data-driven cumulative product schedule outperforms fixed heuristic baselines.

| Schedule | VideoMME | MLVU |
|---|---|---|
| Linear Decay | 59.8 | 64.3 |
| Cosine Decay | 60.5 | 65.1 |
| **Ours (Cumulative Product)** | **62.0** | **67.2** |

**Robustness to DPC-KNN Hyperparameters.** We evaluate the sensitivity of ARTS to the number of nearest neighbors ($K$) and anchors per modality ($M$). As shown in Table 9, performance on VideoMME (Qwen2.5-Omni-3B, 20% retention) is highly stable, exhibiting $< 1.0\%$ variation across the grid. Since larger $K$ and $M$ increase the computational overhead of density estimation and alignment scoring with marginal accuracy gains (e.g., $+0.2$ from $M = 32 \rightarrow 64$), we select $K = 5$ and $M = 32$ to optimally balance accuracy and efficiency.

*Table 9.* Ablation on DPC-KNN hyperparameters ($K$ and $M$). Evaluated on Qwen2.5-Omni-3B at a 20% retention rate on the VideoMME benchmark. The performance variation is minimal ($< 1.0\%$), demonstrating the robustness of the anchor selection process.

| | M = 8 | M = 16 | M = 32 | M = 64 |
|---|---|---|---|---|
| **K = 3** | 61.5 | 61.7 | 61.8 | 61.7 |
| **K = 5** | 61.6 | 61.8 | **62.0** | 61.9 |
| **K = 7** | 61.5 | 61.7 | 61.9 | 61.8 |
| **K = 10** | 61.4 | 61.6 | 61.8 | 61.7 |

## 6.5. Visualization Analysis

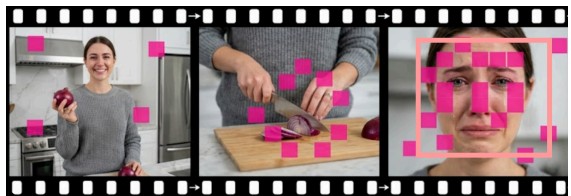

🎵 **Chop, Chop, Chop** (the sound of chopping onions)

*Figure 9.* Case visualization. Given the query "Why is she crying?", OmniFit accurately preserves critical visual tokens (pink patches). Despite the dominant chopping sound, our method successfully localizes both the contextual cause (the chopping action in Frame 2) and the core visual answer (the crying face in Frame 3).

As shown in Fig. 9, OmniFit demonstrates precise semantic alignment even with audio interference. While the audio contains dominant chopping sounds, our method successfully identifies relevant visual tokens, accurately localizing both the chopping action (Frame 2) and the critical crying face (Frame 3) to answer the user query.

## 7. Conclusion

In this work, we proposed *OmniFit*, a novel framework for *token compression* in omnimodal large language models (OmniLLMs). We first identify two motivations: The overlooked layer-wise heterogeneity and the invisible yet critical cross-modal dominance. Based on these findings, we introduced a *layer-adaptive heterogeneity profiling* (LADP) method and a *alignment-rectified token selection* mechanism that enable the model to adaptively compress tokens according to layer-specific heterogeneity while accurately preserving the important portion. Extensive experiments on large-scale benchmarks demonstrate that OmniFit provides significant computational savings without sacrificing performance.

## Acknowledgement

This work was supported by the National Natural Science Foundation of China (Nos. 62525601, 62476018, 62306025) and the Fundamental Research Funds for the Central Universities.

## Impact Statement

This paper presents work whose goal is to advance the field of Machine Learning. There are many potential societal consequences of our work, none which we feel must be specifically highlighted here.

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

# Appendix

This document supplements the main paper as follows:

## A. Theoretical Analysis and Proofs

In this section, we provide the mathematical derivations and proofs supporting the design of Layer-Adaptive Heterogeneity Profiling (LAHP). Specifically focusing on the validity of SVD-based redundancy measurement and the convexity-induced overhead of non-uniform allocation.

### A.1. Theoretical Justification of SVD-based Information Density

In the main paper, we utilize the Effective Rank derived from SVD to quantify the information density of each layer. Here, we justify this metric using the *Eckart-Young-Mirsky Theorem*, which relates the tail singular values to the reconstruction error (i.e., information loss).

**Definition 1 (Token Feature Matrix).** Let $\mathbf{X}^{(l)} \in \mathbb{R}^{N \times d}$ denote the token feature matrix at layer $l$, where $N$ is the sequence length and $d$ is the hidden dimension.

**Theorem 1 (Eckart-Young-Mirsky).** Let the Singular Value Decomposition of $\mathbf{X}^{(l)}$ be $\mathbf{U}\boldsymbol{\Sigma}\mathbf{V}^\top$, where $\boldsymbol{\Sigma} = \mathrm{diag}(\sigma_1, \sigma_2, \ldots, \sigma_r)$ with singular values ordered as $\sigma_1 \geq \sigma_2 \geq \cdots \geq \sigma_r \geq 0$. For any rank $k < r$, the optimal rank-$k$ approximation $\mathbf{X}_k^{(l)}$ that minimizes the reconstruction error under the Frobenius norm is given by the truncated SVD. The approximation error is explicitly bounded by the sum of the squared tail singular values:

$$\min_{\mathrm{rank}(\hat{\mathbf{X}}) \leq k} \|\mathbf{X}^{(l)} - \hat{\mathbf{X}}\|_F^2 = \|\mathbf{X}^{(l)} - \mathbf{X}_k^{(l)}\|_F^2 = \sum_{i=k+1}^{r} \sigma_i^2. \tag{I}$$

*Proof.* The proof follows directly from the unitary invariance of the Frobenius norm. Let $\mathbf{E} = \mathbf{X}^{(l)} - \hat{\mathbf{X}}$. The minimization of $\|\mathbf{E}\|_F^2$ is equivalent to minimizing the sum of squared singular values of the residual matrix. By retaining the largest $k$ singular values, we minimize the residual energy, leaving exactly $\sum_{i=k+1}^{r} \sigma_i^2$. $\qquad\square$

**Proposition 1 (Effective Rank as Redundancy Proxy).** We define the Effective Rank $k_{eff}^{(l)}$ as the minimum dimensionality required to preserve a cumulative energy ratio $\delta$ (e.g., $\delta = 0.9$):

$$k_{eff}^{(l)} = \min\left\{ k \;\middle|\; \frac{\sum_{i=1}^{k} \sigma_i^2}{\sum_{j=1}^{r} \sigma_j^2} > \delta \right\}. \tag{II}$$

*Implication:* If a layer exhibits a low $k_{eff}^{(l)}$, Theorem 1 implies that the tokens can be projected onto a low-dimensional subspace with a bounded information loss of $(1 - \delta)\|\mathbf{X}^{(l)}\|_F^2$. This mathematically certifies that the layer possesses high **linear redundancy** and sparse semantic information, justifying a lower token retention budget. Conversely, a high $k_{eff}^{(l)}$ implies a flat spectrum where most dimensions (tokens) contribute significantly to the total energy, necessitating higher retention.

### A.2. Convexity Analysis of Layer-wise Allocation

In this section, we provide a rigorous proof that for a fixed total token budget, any non-uniform (heterogeneous) allocation strictly increases the computational overhead compared to a uniform allocation. This derivation utilizes the *Cauchy-Schwarz Inequality* to specifically address the quadratic complexity of the self-attention mechanism.

**Computational Cost Model.** The inference cost of a Transformer layer $l$ with $n_l$ tokens is modeled as:

$$\mathcal{C}(n_l) = c_1 n_l + c_2 n_l^2, \tag{III}$$

where $c_1 n_l$ represents linear operations (e.g., FFN, projections) and $c_2 n_l^2$ represents the quadratic attention operations ($c_1, c_2 > 0$).

**Proposition 2 (Optimality of Uniform Allocation).** Let $N_{total}$ be the fixed total token budget distributed across $L$ layers, such that $\sum_{l=1}^{L} n_l = N_{total}$. Let $n_{avg} = N_{total}/L$ be the count for uniform allocation. We assert that:

$$\mathcal{C}_{Hetero} = \sum_{l=1}^{L} \mathcal{C}(n_l) \geq \sum_{l=1}^{L} \mathcal{C}(n_{avg}) = \mathcal{C}_{Uniform}. \tag{IV}$$

*Proof.* The total cost can be expanded as:

$$\sum_{l=1}^{L} \mathcal{C}(n_l) = c_1 \underbrace{\sum_{l=1}^{L} n_l}_{\text{Constant } N_{total}} + c_2 \sum_{l=1}^{L} n_l^2. \tag{V}$$

Since the linear term is invariant under the constraint $\sum n_l = N_{total}$, minimizing the total cost is equivalent to minimizing the quadratic term $S = \sum_{l=1}^{L} n_l^2$.

We apply the *Cauchy-Schwarz Inequality* to the vectors $\mathbf{a} = (n_1, \ldots, n_L)$ and $\mathbf{b} = (1, \ldots, 1)$:

$$\left( \sum_{l=1}^{L} n_l \cdot 1 \right)^2 \leq \left( \sum_{l=1}^{L} n_l^2 \right) \left( \sum_{l=1}^{L} 1^2 \right). \tag{VI}$$

Substituting $\sum n_l = N_{total}$ and $\sum 1^2 = L$:

$$(N_{total})^2 \leq \left( \sum_{l=1}^{L} n_l^2 \right) \cdot L. \tag{VII}$$

Rearranging the inequality yields the lower bound for the sum of squares:

$$\sum_{l=1}^{L} n_l^2 \geq \frac{(N_{total})^2}{L} = L \cdot \left( \frac{N_{total}}{L} \right)^2 = \sum_{l=1}^{L} n_{avg}^2. \tag{VIII}$$

Equality holds if and only if $n_1 = n_2 = \cdots = n_L = n_{avg}$.

Thus, any deviation from uniformity ($n_l \neq n_{avg}$) strictly increases the sum of squares $\sum n_l^2$, thereby increasing the total inference cost. $\qquad\square$

**Corollary (Variance-Induced Overhead).** The relationship derived above can be reformulated to explicitly link the excess cost to the variance of the token distribution. Using the identity $\sum n_l^2 = L(\text{Var}(\mathbf{n}) + n_{avg}^2)$, the cost difference is:

$$\Delta\mathcal{C} = c_2 \left( \sum_{l=1}^{L} n_l^2 - L n_{avg}^2 \right) = c_2 \cdot L \cdot \text{Var}(\mathbf{n}). \tag{IX}$$

This explicitly demonstrates that the computational overhead is proportional to the variance of the layer-wise token budget. To ensure our method remains efficient ($\Delta\mathcal{C} \leq 0$ relative to the baseline), the penalty factor $\xi < 1$ (derived in Eq. 5) is theoretically necessary to counteract this variance term. $\qquad\square$

# B. Hyperparameter Sensitivity Analysis

In this section, we first provide a comprehensive overview of all hyperparameters involved in the OmniFit framework to ensure reproducibility. We then conduct rigorous sensitivity experiments to demonstrate that our method is robust to parameter variations and does not rely on extensive tuning.

## B.1. Overview of Hyperparameters

OmniFit is designed to be parameter-efficient. The framework involves three primary hyperparameters, each governing a specific component of the pipeline:

- **Energy Threshold $\delta$ (Eq. 3):** This parameter determines the strictness of the Effective Rank calculation in *Token Redundancy Profiling*. It represents the cumulative energy ratio preserved after SVD truncation. A higher $\delta$ makes the profiling more conservative (estimating higher ranks), while a lower $\delta$ implies more aggressive redundancy estimation. We set $\delta = 0.9$ by default.

- **Cross-Modal Balance Factor $\lambda$ (Eq. 8):** This is the most critical inference-time hyperparameter in *Alignment-Rectified Token Selection (ARTS)*. It controls the weight of the cross-modal alignment score relative to the intra-modal magnitude. A larger $\lambda$ encourages the model to prioritize tokens that act as "bridges" between modalities (e.g., visual tokens aligned with audio anchors). We set $\lambda = 1.5$ by default.

- **Calibration Sample Size $N_{calib}$:** This parameter defines the number of samples used during the offline *Phase I: Calibration*. It balances the precision of layer-wise profiling against the one-time preprocessing cost. Our main experiments use $N_{calib} = 1024$.

## B.2. Sensitivity Experiments

To verify the robustness of OmniFit, we evaluate the impact of varying these hyperparameters on the Qwen2.5-Omni-3B model under a 20% token retention budget ($\mu = 0.2$).

**Impact of cross-modal balance factor $\lambda$.** We vary $\lambda$ within the range $[0.5, 3.0]$ to assess how the trade-off between intra-modal saliency and cross-modal alignment affects performance.

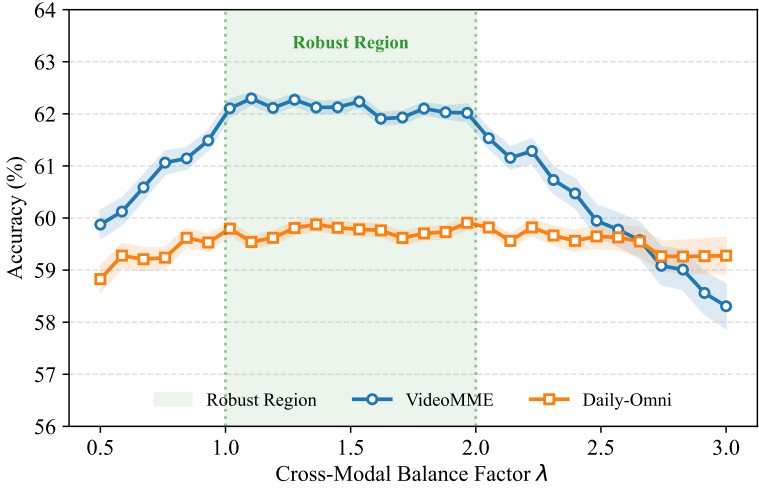

*Figure I.* **Sensitivity to $\lambda$.** The performance on VideoMME and Daily-Omni remains stable for $\lambda \in [1.0, 2.0]$.

As illustrated in Fig. I, the model performance is highly stable.

- **Under-weighting ($\lambda < 1.0$):** Performance drops slightly because the selection reverts to being dominated by intra-modal norms, failing to capture silent but aligned visual cues (Observation iii).

- **Optimal Range** ($1.0 \leq \lambda \leq 2.0$)**:** The accuracy fluctuates by less than $0.3\%$, indicating a wide "sweet spot" for this parameter.

- **Over-weighting** ($\lambda > 2.5$)**:** Performance begins to degrade as the model may over-prioritize weak cross-modal signals at the expense of fundamental visual features.

Table 7 demonstrates that OmniFit converges rapidly. Using only **256 samples** yields performance nearly identical to using 1024 or 2048 samples. This efficiency implies that the layer-wise heterogeneity patterns (Observations i & ii) are intrinsic and consistent properties of the model, which can be captured with very little data. This ensures that OmniFit can be deployed to new models with negligible calibration time (*e.g.*, $< 5$ minutes on a single GPU for 256 samples).

**Sensitivity to Energy Threshold** $\delta$. The energy threshold $\delta$ in Eq. (3) governs the aggressiveness of the Token Redundancy Profiling. It determines how much spectral energy must be preserved when estimating the effective rank.

- A lower $\delta$ (e.g., 0.8) leads to aggressive rank estimation, potentially discarding subtle but critical information in shallow layers.

- A higher $\delta$ (e.g., 0.99) forces the estimator to preserve nearly full variance, causing the layer-wise budget allocation to degenerate into a uniform distribution, which we have proven to be suboptimal (Observation i).

To verify this, we evaluate the performance on VideoMME and Daily-Omni with $\delta \in \{0.8, 0.85, 0.9, 0.95, 0.99\}$.

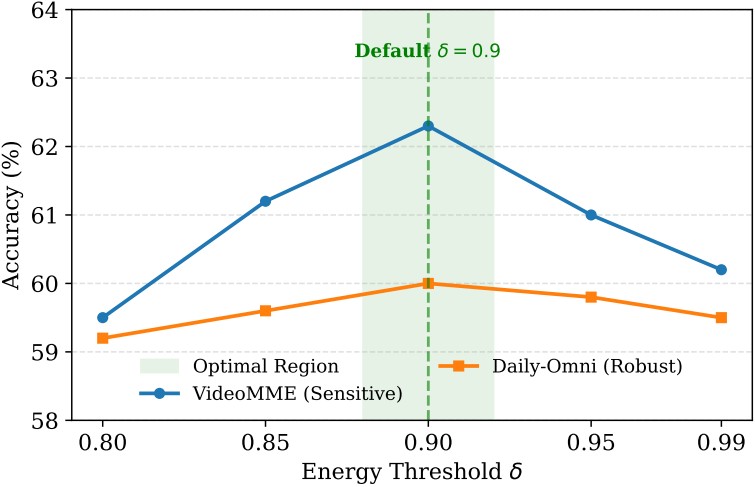

*Figure II.* **Impact of Energy Threshold** $\delta$**.** The performance peaks at $\delta = 0.9$. Extreme values lead to degradation: $\delta = 0.8$ suffers from information loss, while $\delta = 0.99$ suffers from the inefficiency of uniform allocation.

As shown in Figure II, setting $\delta = 0.9$ consistently yields the best results. The performance drop at $\delta = 0.99$ strongly supports our motivation that *layer-adaptive heterogeneity* is superior to uniform strategies. Consequently, we fix $\delta = 0.9$ as the standard setting.

## C. Computational Complexity Analysis

In this section, we provide a theoretical analysis of the time and space complexity of OmniFit, distinguishing between the offline calibration phase and the online inference phase. We demonstrate that our method introduces negligible overhead while significantly reducing the dominant quadratic cost of the self-attention mechanism.

### C.1. Notations

Let $N$ denote the sequence length, $d$ the hidden dimension, $L$ the number of Transformer layers, and $N_{calib}$ the number of samples used for calibration. Let $r^{(l)} \in (0, 1]$ be the token retention ratio at layer $l$, and $M$ be the number of global anchors per modality (where $M \ll N$).

## C.2. Phase I: Offline Calibration (LAHP)

The calibration phase involves two primary computations: Token Redundancy Profiling (TRP) and Modality Preference Profiling (MPP).

- **TRP (SVD):** Computing the Singular Value Decomposition (SVD) for a hidden state matrix $\mathbf{X} \in \mathbb{R}^{N \times d}$ has a complexity of $\mathcal{O}(Nd^2)$. Across $L$ layers and $N_{calib}$ samples, the total cost is $\mathcal{O}(N_{calib} \cdot L \cdot Nd^2)$.

- **MPP (Attention Aggregation):** Aggregating attention scores requires summing over the attention map, costing $\mathcal{O}(N^2)$ per layer.

**Total Offline Cost:** $\mathcal{O}(N_{calib} \cdot L \cdot (Nd^2 + N^2))$. Since this is a **one-time offline process**, its amortized cost during inference is zero. As shown in Fig. 7, calibrating a 7B model takes less than 30 minutes.

## C.3. Phase II: Online Inference (ARTS)

During inference, the computational bottleneck of a standard Transformer is the self-attention mechanism, which scales quadratically as $\mathcal{O}(N^2d)$. OmniFit replaces this with a pruned attention mechanism guided by ARTS. Crucially, unlike prior methods that re-evaluate token importance at every layer, OmniFit computes importance scores **only once** immediately after encoding (as detailed in Alg. 2), significantly reducing the selection overhead.

### C.3.1. OVERHEAD OF TOKEN SELECTION

We analyze the complexity of the ARTS mechanism, distinguishing between the one-time global calculation and the layer-wise pruning operations:

1. **Global Importance Scoring (Computed Once):** Immediately after the encoder, we calculate the $S_{intra}$ ($L_2$ norm) and $S_{cross}$ (Cosine Similarity with $M$ anchors) for all tokens. This involves a single matrix multiplication of $\mathcal{O}(N \cdot M \cdot d)$. Since the number of anchors is small and constant ($M \ll N$), this step has a linear complexity of $\mathcal{O}(Nd)$.

2. **Layer-wise Pruning (Per Layer):** At each layer $l$, we simply select the top-$k$ tokens based on the *pre-computed* global scores. Using algorithms like QuickSelect, this requires only $\mathcal{O}(N)$ operations per layer, independent of the hidden dimension $d$.

**Total Selection Overhead:** The aggregate selection cost across $L$ layers is $\mathcal{O}(Nd + L \cdot N)$. Since $d$ is typically large (e.g., 4096), the term $Nd$ dominates, making the total overhead effectively $\mathcal{O}(Nd)$. This is fundamentally more efficient than per-layer selection methods, which incur $\mathcal{O}(L \cdot Nd)$.

### C.3.2. OVERALL INFERENCE COMPLEXITY

By combining the efficient selection with sparse attention, the total inference complexity is:

- **Standard Inference:** $\mathcal{O}(L \cdot N^2d)$.

- **OmniFit Inference:** $\mathcal{O}(\underbrace{Nd}_{\text{Global Score}} + L \cdot (\underbrace{N}_{\text{Selection}} + \underbrace{(r^{(l)}N)^2d}_{\text{Sparse Attn}}))$.

Given that $r^{(l)} \approx 0.2$ and the selection cost is negligible, the dominant quadratic term is reduced by approximately $96\%$, leading to substantial end-to-end speedups.

## C.4. Complexity Comparison

We summarize the theoretical complexity of OmniFit against the Full Attention baseline and existing pruning methods in Table I.

*Table I.* **Comparison of Computational Complexity.** OmniFit achieves the lowest selection overhead by computing scores once globally, whereas baselines incur repeated calculation costs at every layer.

| Method | Total Selection Cost | Attention Cost |
|---|---|---|
| Full Attention | - | $\mathcal{O}(L \cdot N^2 d)$ |
| OmniZip (Tao et al., 2025b) | $\mathcal{O}(L \cdot Nd)$ | $\mathcal{O}(L \cdot (rN)^2 d)$ |
| EchoingPixels (Gong et al., 2025) | $\mathcal{O}(L \cdot N^2 d)$ | $\mathcal{O}(L \cdot (rN)^2 d)$ |
| **OmniFit (Ours)** | $\boldsymbol{\mathcal{O}(Nd)}$ | $\boldsymbol{\mathcal{O}(L \cdot (rN)^2 d)}$ |

## C.5. Memory Complexity Analysis

In addition to computational speedups, OmniFit significantly reduces memory footprint, particularly for the Key-Value (KV) cache. In standard LLMs, the KV cache grows linearly with sequence length: $\mathcal{O}(L \cdot N \cdot d)$. OmniFit actively prunes tokens at each layer, preventing the KV cache from accumulating redundant information. The memory complexity for the KV cache becomes $\mathcal{O}(\sum_{l=1}^{L} r^{(l)} \cdot N \cdot d)$. For a target compression ratio $\mu$, the memory usage is approximately $\mu\times$ that of the baseline, enabling the deployment of OmniLLMs on memory-constrained edge devices.

# D. Detailed Algorithms

In this section, we provide the pseudocode for the two phases of OmniFit: the offline calibration phase (Layer-Adaptive Heterogeneity Profiling) and the online inference phase (Alignment-Rectified Token Selection).

## D.1. Phase I: Offline Calibration

Algorithm 1 details the process of *Layer-Adaptive Heterogeneity Profiling (LAHP)*. This phase consumes a small calibration dataset to profile the depth-dependent redundancy (via SVD) and layer-wise modality preferences (via Attention Aggregation). The output profiles are fixed and cached for inference.

## D.2. Phase II: Online Inference

Algorithm 2 describes the *Alignment-Rectified Token Selection (ARTS)* process during inference. Note that the global anchors $\mathcal{A}$ are pre-computed using DPC-KNN clustering on the training set (as described in Sec. 5.2).

# E. More Results on Efficiency

In this section, we provide the detailed numerical results corresponding to the efficiency analysis presented in Figure 8 of the main text. We report the Time to First Token (TTFT) for the prefilling stage and the Time Per Output Token (TPOT) for the decoding stage across different settings.

## E.1. Detailed Latency Comparison

Tables II and III present the exact latency measurements for Qwen2.5-Omni-7B and Qwen3-Omni-30B-A3B-Instruct, respectively. We compare **OmniFit** against the original Full Attention baseline and state-of-the-art pruning methods (OmniZip).

---

**Algorithm 1** Offline Calibration (LAHP)

---

1: **Input:** Calibration dataset $\mathcal{D}_{cal}$, Pre-trained OmniLLM $\mathcal{M}$ with $L$ layers, Global compression ratio $\mu$, Energy threshold $\delta$.
2: **Output:** Layer-wise retention rates $\mathbf{r} = \{r^{(1)}, \dots, r^{(L)}\}$, Modality preference scores $\boldsymbol{\rho} = \{\rho^{(1)}, \dots, \rho^{(L)}\}$.
3: Initialize accumulators for ranks $\Psi^{(l)} \leftarrow 0$ and attention scores $A_{sum}^{(l)} \leftarrow 0$.
4: **for** each sample $X$ in $\mathcal{D}_{cal}$ **do**
5:     Forward pass $X$ through $\mathcal{M}$ to get hidden states $H^{(l)}$ and attention maps.
6:     **for** $l = 1$ to $L$ **do**
7:         Compute SVD: $U, \Sigma, V^T \leftarrow \text{SVD}(H^{(l)})$.         ▷ **Token Redundancy Profiling (TRP)**
8:         Calculate effective rank $k_{eff}^{(l)}$ such that energy ratio $> \delta$ (Eq. 3).
9:         Update accumulator: $\Psi^{(l)} \leftarrow \Psi^{(l)} + k_{eff}^{(l)}$.
10:        Extract attention weights corresponding to each modality $m$.     ▷ **Modality Preference Profiling (MPP)**
11:        Update accumulator: $A_{sum}^{(l)} \leftarrow A_{sum}^{(l)} + \text{SumAttention}(m)$.
12:     **end for**
13: **end for**
14: **Profile Aggregation:**
15: **for** $l = 1$ to $L$ **do**
16:     Average rank: $\bar{\Psi}^{(l)} \leftarrow \Psi^{(l)} / |\mathcal{D}_{cal}|$.
17:     Normalize attention to get preference scores $\rho_m^{(l)}$ (Eq. 6).
18:     Calculate unscaled profile based on rank decay: $\hat{r}^{(l)} \propto \bar{\Psi}^{(l)}$.     ▷ **Solve for Retention Rates**
19:     Solve quadratic equation (Eq. 5) for penalty factor $\xi$ to satisfy $\sum \mathcal{C}(r^{(l)}N) \leq \mathcal{C}_{Uniform}$.
20:     Set final retention rate: $r^{(l)} \leftarrow \xi \cdot \hat{r}^{(l)}$.
21: **end for**
22: **return** $\mathbf{r}, \boldsymbol{\rho}$

---

---

**Algorithm 2** Online Inference with ARTS

---

1: **Input:** Input sequence $X$, Calibration Profiles $\mathbf{r}$, $\boldsymbol{\rho}$, Balance factor $\lambda$.
2: **Output:** Generated Response $Y$.
3: **Step 0: Encoding & Initialization**
4: Encode inputs: $H^{(0)} \leftarrow \text{Encoder}(X)$.
5: **for** each modality $m \in \{Audio, Vision\}$ **do**
6:    $\mathcal{A}_m \leftarrow \text{DPC-KNN}(H_m^{(0)})$.                                             ▷ Instance-specific anchors
7: **end for**
8: **Step 1: Global Importance Scoring (Computed Once)**
9:                                  ▷ Calculate importance scores $\mathcal{S}$ for all tokens using initial features $H^{(0)}$
10: **for** each token $h_i$ in modality $m$ **do**
11:    $S_{intra} \leftarrow \|h_i\|_2$
12:    $S_{cross} \leftarrow \max_{a_k \in \mathcal{A}_{\neg m}} \text{CosineSim}(h_i, a_k)$
13:                       ▷ Note: We use the average preference $\bar{\rho}$ or first layer's $\rho^{(1)}$ for global scoring
14:    $S_i \leftarrow S_{intra} + \lambda \cdot \bar{\rho}_{\neg m} \cdot \text{ReLU}(S_{cross})$
15: **end for**
16: **Step 2: Layer-wise Pruning & Execution**
17: Let $\mathcal{I}_{active}$ be the set of indices of currently active tokens (initially all).
18: **for** $l = 1$ to $L$ **do**
19:    Retrieve $r^{(l)}$ from calibration profile.                           ▷ 2.1 Determine Budget
20:    Calculate target count $k_m = r_m^{(l)} \cdot N_m$ for each modality.
21:    **for** each modality $m$ **do**
22:       Identify top-$k_m$ indices in $\mathcal{I}_{active}$ based on scores $\{S_i\}$.     ▷ 2.2 Pruning (Based on pre-computed $\mathcal{S}_i$)
23:       Update $\mathcal{I}_{active}$ to keep only selected tokens.
24:    **end for**
25:    $H^{(l)} \leftarrow \text{SelfAttention}(H^{(l-1)}[\mathcal{I}_{active}]) + \text{FFN}(\dots)$         ▷ 2.3 Transformer Forward
26: **end for**
27: **Step 3: Decoding**
28: **return** Generate next token from $H^{(L)}$.

---

*Table II.* **Efficiency Breakdown on Qwen2.5-Omni-7B.** We report TTFT (ms) for varying sequence lengths and TPOT (ms/token) for varying batch sizes. OmniFit consistently achieves the lowest latency, with speedups reaching up to **2.20**× in long-context prefilling.

| Stage | Setting | Origin | OmniZip | OmniFit (Ours) | Speedup |
|---|---|---|---|---|---|
| **Prefilling** (TTFT, ms) | Length=512 | 280 | 225 | **231** | 1.21× |
| | Length=1024 | 459 | 311 | **295** | 1.55× |
| | Length=2048 | 855 | 522 | **387** | **2.20**× |
| **Decoding** (TPOT, ms/token) | Batch=8 | 12.4 | 11.8 | **11.8** | 1.05× |
| | Batch=16 | 18.2 | 17.1 | **16.5** | 1.10× |
| | Batch=32 | 32.5 | 28.9 | **27.0** | 1.20× |

## F. Visualization of Token Survival

To strictly quantify the compression progression on the 28-layer Qwen2.5-Omni-7B, we visualize the Cumulative Token Survival Rate in Fig. III(a). This metric represents the percentage of initial tokens remaining after each layer, exhibiting a strictly monotonic decrease.

The survival curves in Figure III(a) clearly illustrate the three-stage processing pipeline:

1. **Shallow Layers (1-8):** The Vision curve remains high ($> 60\%$), confirming that the model prioritizes preserving visual details during the initial feature extraction phase.

*Table III.* **Efficiency Breakdown on Qwen3-Omni-30B-A3B-Instruct.** Comparison of latency metrics. OmniFit demonstrates superior scalability, particularly in the prefilling stage for longer sequences.

| Stage | Setting | Origin | OmniZip | OmniFit (Ours) | Speedup |
|---|---|---|---|---|---|
| **Prefilling** (TTFT, ms) | Length=256 | 248 | 240 | **236** | 1.05× |
| | Length=512 | 420 | 350 | **315** | 1.33× |
| | Length=1024 | 1151 | 681 | **497** | **2.31×** |
| **Decoding** (TPOT, ms/token) | Batch=1 | 50.1 | 49.5 | **49.1** | 1.02× |
| | Batch=2 | 68.5 | 60.2 | **58.3** | 1.17× |
| | Batch=4 | 105.0 | 92.0 | **75.5** | 1.39× |

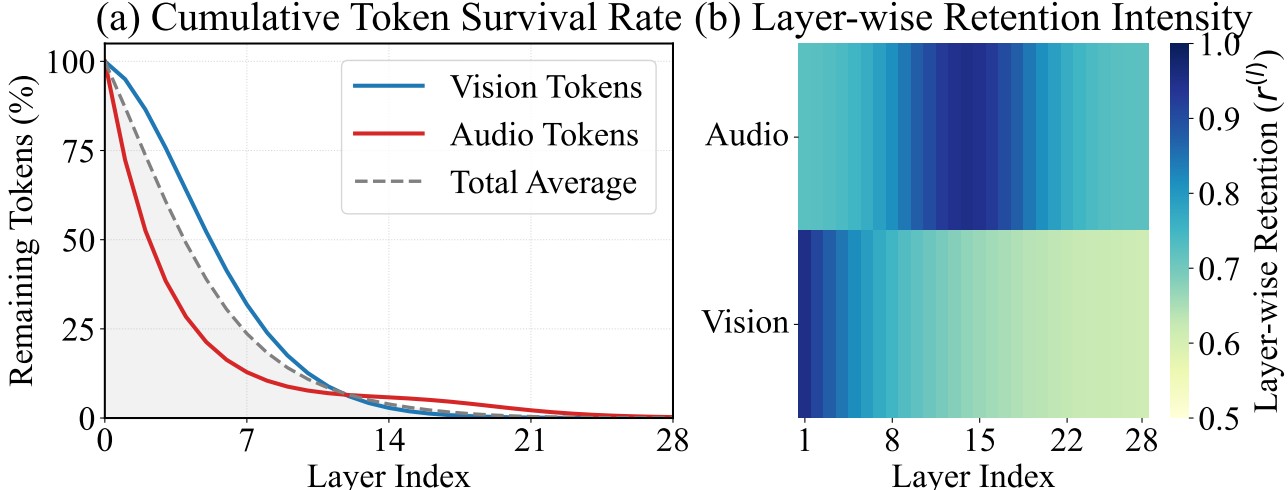

*Figure III.* Token Survival Analysis on 28 Layers. (a) The monotonic decay of token quantity. The **Vision** curve (Blue) drops steadily after the initial perception layers. The **Audio** curve (Red) exhibits a distinct "plateau" in the middle layers (9-19), indicating a "protection zone" where semantic audio tokens are preserved for cross-modal reasoning. (b) The heatmap confirms the retention intensity at each stage.

2. **Middle Layers (9-19):** The Audio curve flattens (slower decay), creating a visible gap between Audio and Vision. This aligns with the "Semantic Fusion" phase, where audio instructions serve as critical anchors for reasoning, requiring higher retention.

3. **Deep Layers (20-28):** Both curves drop rapidly as the redundancy profiler detects high abstraction, reducing the sequence to only the most essential tokens ($< 15\%$) for the final prediction.

## G. Ablation: Flexibility of OmniFit (Pruning vs. Merging)

In this ablation study, we verify the flexibility of the OmniFit framework by demonstrating that once the essential tokens are identified via the ARTS algorithm, the framework can seamlessly accommodate either **Token Merging** (soft aggregation) or **Token Pruning** (hard selection) strategies. As shown in Tab. IV, while the default merging strategy achieves the highest accuracy of 62.9% by preserving residual context through aggregation, the pruning variant remains remarkably robust. Specifically, the Pruning variant trades a marginal 0.3% accuracy drop (62.5%) for a significant 11.8% reduction in inference latency (from 288ms to 251ms). This trade-off underscores the precision of the OmniFit importance scoring mechanism, which captures core semantic tokens so accurately that even their aggressive isolation via hard pruning maintains superior understanding performance. Consequently, the Pruning variant serves as an ideal high-efficiency alternative for latency-critical edge deployments, where the substantial speedup justifies the minor sacrifice in precision.

**Analytical Insight on Speedup.** The $\sim$12% latency advantage of *OmniFit-Prune* over *OmniFit-Merge* stems from three hardware-level factors: (1) **Algorithmic Simplicity:** Pruning eliminates the $O(N^2)$ similarity matching required by merging, replacing it with a $O(N)$ top-$k$ selection. (2) **Memory Efficiency:** Pruning involves simple index-based slicing, which

*Table IV*. **OmniFit Flexibility: Merging vs. Pruning under Various Retention Rates.** Evaluated on Qwen2.5-Omni-3B using VideoMME. As the retention rate $\rho$ decreases, the Merging strategy demonstrates superior information preservation, while the pruning variant provides a consistent 12-15% latency advantage across all settings.

| Retention ($\mu$) | Strategy | VideoMME (%) | Latency (ms) | $\triangle$ Acc. | Speedup |
|---|---|---|---|---|---|
| 40% | OmniFit-Merge | **62.9** | 288 | - | - |
| | OmniFit-Prune | 62.5 | **251** | **-0.4** | **+12.8%** |
| 30% | OmniFit-Merge | **62.5** | 245 | - | - |
| | OmniFit-Prune | 61.8 | **216** | **-0.7** | **+11.8%** |
| 20% | OmniFit-Merge | **62.0** | 212 | - | - |
| | OmniFit-Prune | 60.5 | **184** | **-1.5** | **+13.2%** |

benefits from coalesced memory access, whereas merging requires scattered read-writes for feature aggregation. (3) **Reduced Kernel Overhead:** Merging introduces additional CUDA kernels in each layer for token alignment and fusion. In our 28-layer architecture, the elimination of these kernels significantly reduces the total launch latency, making Pruning a highly efficient choice for real-time inference.

# H. Ablation: Frequency of Token Selection

In this ablation study, we evaluate the trade-off between the dynamicity of token selection and total inference overhead by comparing a **Static Selection** strategy (calculating importance only once at the input layer) against our default **Dynamic Selection** (per-layer scoring). As shown in Table V, while dynamic selection at every layer yields the highest accuracy (63.0%) by capturing feature drift in deeper representations, the static variant remains remarkably robust. Specifically, the *Once* variant achieves a competitive 61.5% accuracy while providing a 15.1% speedup, reducing latency to 208ms. This suggests that the initial modality alignment and semantic anchors identified by ARTS are highly representative throughout the entire forward pass. These results demonstrate that OmniFit can be seamlessly scaled: for high-fidelity reasoning, per-layer selection is preferred, whereas for latency-critical real-time applications, a single initial selection offers an optimal balance between throughput and precision.

*Table V*. Comparison of Token Selection Frequency. Evaluated on Qwen2.5-Omni-3B with a 30% retention rate. Dynamic selection at every layer provides the highest accuracy but introduces non-negligible latency overhead due to repeated scoring and indexing.

| Frequency | Selection Logic | VideoMME (%) | Latency (ms) | Efficiency Gain |
|---|---|---|---|---|
| **Once (adopt in main paper)** | Static (Input only) | 62.5 | **208** | **+15.1%** |
| **Every Layer** | Dynamic (Per-layer) | **62.8** | 245 | - |
| *Gap* | - | **-0.4%** | **-37ms** | - |

# I. Merge Strategy

**Step 0: Anchor Selection and Token Importance.** We first compute the token importance scores $\mathcal{S}_i$ and perform DPC-KNN to select anchors directly from the input tokens of each modality. The anchors are selected by identifying the most representative tokens in the input using DPC-KNN, forming a set of anchors $\mathcal{I}_{keep}$. Then we calculate the importance scores $\mathcal{S}_i$ of each token, and remain fixed throughout the model layers. Thus, the anchor set $\mathcal{I}_{keep}$, and the importance scores $\mathcal{S}_i$ are calculated only once, and are then used for the subsequent layers during inference.

**Step 1: Neighbor Assignment to Fixed Anchors (Per-Layer).** After computing the importance scores $\mathcal{S}_i$ for each token, we assign each dropped token $r \in \mathcal{I}_{drop}$ to its nearest anchor in $\mathcal{I}_{keep}$:

$$j^*(r) = \arg \max_{j \in \mathcal{I}_{keep}} \cos(\mathbf{x}_r, \mathbf{x}_j), \tag{X}$$

where $\mathbf{x}_r$ and $\mathbf{x}_j$ represent the token embeddings of the dropped token $r$ and the anchor token $j$, respectively. The assigned anchor neighbors are denoted as $\mathcal{N}(j)$, which are the set of dropped tokens $r$ assigned to anchor $j$.

**Step 2: ARTS-Weighted Aggregation (Per-Layer).** At each layer, we apply a weighted aggregation of the neighbors to

refine the representation of the retained tokens. The aggregation is done using the precomputed importance scores $\mathcal{S}_i$ as the weighting factor:

$$\mathbf{x}'_j = \frac{\mathcal{S}_j \cdot \mathbf{x}_j + \sum_{r \in \mathcal{N}(j)} \mathcal{S}_r \cdot \mathbf{x}_r}{\mathcal{S}_j + \sum_{r \in \mathcal{N}(j)} \mathcal{S}_r}, \tag{XI}$$

where $\mathcal{S}_j$ and $\mathcal{S}_r$ are the importance scores of the anchor token $j$ and the dropped token $r$, respectively.

**Implementation Note.** The DPC-KNN and token importance calculation is applied *once* immediately after the encoder, producing a compressed sequence $\{\mathbf{x}'_j\}_{j \in \mathcal{I}_{keep}}$ that is reused by all subsequent layers without any per-layer rescoring. Each layer, we conduct token merging.

**Complexity.** The assignment requires computing similarities between dropped and retained tokens, with a naive complexity of $O(|\mathcal{I}_{drop}| \cdot |\mathcal{I}_{keep}|)$; we use top-$R$ anchors with small $R$ (default $R{=}1$) and batched matrix multiplication (or ANN) to reduce overhead in practice. For each layer, merge only requires scatter-based aggregation:

$$O(|\mathcal{D}^{(l)}| \cdot R \cdot d)$$

where $\mathcal{D}^{(l)}$ denotes the set of dropped tokens at layer $l$, $R$ is the number of anchor candidates, and $d$ is the dimensionality of each token representation.

**Remark.** Although merging is applied at every layer to enable progressive length reduction, both token importance scores and the sparse anchor assignment candidates are computed once on encoder outputs. Each layer only performs scatter-based aggregation using the precomputed candidates, without any per-layer similarity search or score recomputation.

