# OpenReview forum: "OmniFit: Bridging Modalities via Layer-Adaptive Token Compression for Omnimodal Large Language Models"
_ICML.cc/2026/Conference — ICML 2026 spotlight_

### Official Review · Reviewer_6x5t · 2026-03-06

**Soundness:** 3
**Presentation:** 3
**Significance:** 3
**Originality:** 3
**Overall Recommendation:** 5
**Confidence:** 4

**Summary:**

The paper proposes OmniFit, a training-free token compression framework for omni-modal large language models that process joint video, audio, and text inputs. The key idea is to decouple offline profiling from online inference: Layer-Adaptive Heterogeneity Profiling estimates layer-wise redundancy and modality preference to allocate retention budgets, while Alignment-Rectified Token Selection selects informative tokens based on cross-modal alignment with global semantic anchors. Experiments on three OmniLLM families (Qwen2.5-Omni, OmniVinci, and Qwen3-Omni) across ten benchmarks show that OmniFit retains around 98% of the original performance while using only 20% of the tokens, achieving up to 2.3× inference speedup and substantial memory reduction. The method consistently outperforms existing token compression approaches such as OmniZip, FastVid, DyCoKe, and EchoingPixels across different compression regimes.

**Compliance With Llm Reviewing Policy:**

Affirmed.

**Final Justification:**

I thank the authors for the discussion throughout the discussion process; it helped clarify understanding certain points that I missed earlier. Although the authors addressed all my questions regarding the framing and method detail of the paper, I think the paper can benefit from our discussion in the paper; resulting in increasing my rating while understanding and confidence in the paper.

**Key Questions For Authors:**

1. How were FastVid and DyCoKe adapted to the omni-modal setting used in this work? In particular, it would be helpful to clarify which modalities are pruned at different layers and how audio and text tokens are handled during compression. Providing sufficient implementation details would improve reproducibility and help ensure that the comparisons reported in Table 1 reflect competitively implemented baseline.

2. Can you describe how OmniFit behaves in a true streaming setup (e.g., user audio/video chunks arriving continuously, incremental decoding)? Did you run any preliminary experiments in such a setting, and if so, how does latency vs accuracy compare to OmniZip or other baselines? Clarifying this would strengthen the real-time motivation in Section 1.

3. In a streaming or temporally evolving setting, the importance of tokens may change as new multimodal context becomes available. How stable is the token selection strategy of OmniFit under such dynamic inputs? For example, could early token pruning remove information that becomes relevant later in the sequence?

**Limitations:**

The paper would benefit from a clearer discussion of its limitations. In particular, the reliance on an offline calibration stage may introduce additional preprocessing overhead in large-scale deployments. It would also be helpful to discuss potential risks of aggressive token compression for tasks requiring fine-grained temporal reasoning or complex cross-modal interactions. Finally, the current evaluation focuses on static inference settings, and the behavior of the method in streaming or real-time multimodal scenarios remains unclear.

**Strengths And Weaknesses:**

Strengths：
1. The paper addresses the important problem of reducing the computational cost of omni-modal large language models, which process long sequences from video, audio, and text. Improving efficiency without retraining is practically valuable for real-world deployment.
2. The method is motivated by two empirical observations: layer-wise heterogeneity in token redundancy and varying modality preference across layers. These insights provide a reasonable basis for the proposed Layer-Adaptive Heterogeneity Profiling and Alignment-Rectified Token Selection mechanisms.
3. OmniFit operates without modifying or retraining the backbone model, making it applicable to existing OmniLLM systems with minimal engineering overhead.
4. Experiments across three OmniLLM families and multiple benchmarks demonstrate that the method maintains high performance under aggressive compression, achieving substantial speedup and memory reduction compared to prior token compression methods.

Weaknesses:
1. While the proposed framework combines several components (layer-wise profiling and alignment-based selection), the overall idea is closely related to prior work on token pruning, token merging, and modality-aware compression. The paper could better clarify what fundamentally differentiates OmniFit from these existing approaches.
2. The method relies on an offline profiling stage to estimate redundancy and modality preferences. The paper does not clearly quantify the computational overhead of this calibration step or discuss its scalability to very large models or datasets.
3. Although the results demonstrate strong average performance retention, the paper provides limited analysis of scenarios where aggressive token compression might degrade multimodal reasoning, particularly in tasks requiring fine-grained temporal or cross-modal interactions.
4. Section 5.2 mentions using DPC-KNN to distill tokens into anchors, but important hyperparameters (K, number of centers per modality, effect of this choice) are not specified in the main paper. The method’s performance may be sensitive to poor anchor clustering but this is not analyzed.

---

> ### Author Rebuttal · Authors · 2026-03-31
>
> We thank Reviewer 4 for the thorough evaluation. We address each concern below.
>
> **W1: Novelty differentiation from prior work.**
>
> We respectfully highlight three fundamental differences:
>
> (1) *Problem scope*: Prior methods (ToMe, FastVid, DyCoke) operate within a single modality. OmniFit is the first training-free framework jointly addressing layer-wise heterogeneity AND cross-modal alignment for omni-modal (audio+video+text) models.
>
> (2) *Layer-adaptive profiling*: Unlike methods that apply uniform compression across layers, LAHP derives layer-specific retention budgets via SVD-based redundancy analysis and modality preference profiling. The penalty factor ξ (Eq. 5) provides a closed-form guarantee that non-uniform allocation remains computationally efficient — this theoretical contribution is absent in prior work.
>
> (3) *Selection complexity*: EchoingPixels requires O(LN²d); OmniZip requires O(LNd) per-layer scoring. OmniFit scores once globally at O(Nd) using pre-computed anchors, achieving 27.8–42× speedup (Fig. 7 right) — a fundamentally different complexity class for the selection mechanism.
>
> **W2: Calibration overhead and scalability.**
>
> The one-time offline calibration takes ~15 min (3B), ~25 min (7B), ~55 min (30B MoE) on 8×H800 GPUs (Fig. 7 left). Only 256 samples suffice (Appendix Tab.), reducing cost to <5 min for dense models. Complexity is O(N_calib·L·(Nd²+N²)), growing linearly with L, tractable for 100B+. The output is a lightweight cached config — no re-calibration unless the model changes.
>
> **W3: Aggressive compression degrading multimodal reasoning.**
>
> Table 2 provides direct evidence. At 5% retention, OmniFit still outperforms training-based EchoingPixels on VideoMME (55.9 vs 55.7), WorldSense (41.1 vs 40.9), and Daily-Omni (54.1 vs 52.8). At 10%, OmniFit retains 93.6% of baseline performance (58.8/62.8).
>
> The margins between methods narrow at lower ratios: OmniFit leads by 2.2 points at 20% but 0.2 at 5% on VideoMME. We hypothesize that as more tokens are removed, the inherent information loss plays an increasingly larger role relative to the selection strategy. OmniFit's advantage is most pronounced at moderate compression (20–30%), where layer-adaptive allocation can meaningfully differentiate which tokens to preserve.
>
> Regarding failure modes: degradation is most noticeable on tasks requiring fine-grained temporal reasoning (e.g., sequential event ordering in MLVU), where aggressive pruning may remove temporally sparse but semantically critical frames. We will add this discussion in the revision.
>
> **W4: DPC-KNN hyperparameters.**
>
> K=5, M=32 anchors per modality. Robustness verified on Qwen2.5-Omni-3B at 20% retention:
>
> | VideoMME | M=8  | M=16 | M=32     | M=64 |
> | -------- | ---- | ---- | -------- | ---- |
> | K=3      | 61.5 | 61.7 | 61.8     | 61.7 |
> | K=5      | 61.6 | 61.8 | **62.0** | 61.9 |
> | K=7      | 61.5 | 61.7 | 61.9     | 61.8 |
> | K=10     | 61.4 | 61.6 | 61.8     | 61.7 |
>
> <1.0% variation across the grid, confirming robustness. We select K=5, M=32 because larger values increase anchor extraction cost (density scales with K; scoring in Eq. 6 scales with M) while accuracy gains are marginal (M=32→64: +0.2). This balances accuracy and efficiency. We will include this table in the revision.
>
> **Q1: Baseline adaptation (FastVid, DyCoke).**
>
> FastVid's visual pruning was applied independently to both video and audio sequences using each modality's encoder features for importance scoring. DyCoke's hybrid strategy was extended analogously — per-layer per-modality pruning during prefilling, with the KV cache inheriting the pruned set. Text tokens are preserved in full for all methods. We will include these implementation details in the revision.
>
> **Q2/Q3: Streaming behavior and dynamic inputs.**
>
> OmniFit's token selection can support online dynamic updates. Appendix Sec. H (Table 5) shows that dynamic per-layer re-selection adds only 37ms overhead (245ms vs. 208ms) on Qwen2.5-Omni-3B. In a streaming scenario, only new incoming chunks need re-scoring, so the overhead would be substantially lower than this full-reselection cost, remaining within real-time latency bounds.
>
> We acknowledge that a thorough streaming evaluation with incremental anchor update algorithms (e.g., dynamic DPC-KNN that incrementally updates cluster centers as new tokens arrive) represents a promising future direction. We will add this discussion to the limitations section.

---

> > ### Author Rebuttal · Reviewer_6x5t · 2026-04-02
> >
> > I am pleased to see the authors detailed response, my concerns have been adequately addressed. Please include our discussion into the revision. Based on this, I will update my score accordingly.

---

### Official Review · Reviewer_xkEa · 2026-03-12

**Soundness:** 4
**Presentation:** 3
**Significance:** 3
**Originality:** 3
**Overall Recommendation:** 5
**Confidence:** 3

**Summary:**

This paper proposes a training-free token compression framework for efficient Omni-modal LLM. Through layer-wise profiling and following token selection, it highly reduces the token costs while not hurting the accuracy.

**Compliance With Llm Reviewing Policy:**

Affirmed.

**Final Justification:**

As my original evaluation is positive based on the insightful motivation and design, I will keep my current score.

**Key Questions For Authors:**

1. What does the 'Avg. (%)' mean? This does not seem as the average of the accuracy or token costs.

2. In several cases, OminZip works better than the proposed OmniFit. Are there any analysis on these results?

3. In the appendix, the retention intensity seems quite monotonous for each modality (increasing-decreasing or decreasing rather than dynamically changing). Any explanation on this?

**Limitations:**

No. It is recommended to explain the cases where and why the proposed OmniFit works poorer than the existing approaches. Furthermore, due to the layer-wise compression, it might incur higher complexity.

**Strengths And Weaknesses:**

< Strengths >

1. Strong Motivation. Although OmniLLMs are powerful, memory-efficient multi-modal model on a shared Transformer architecture, (as this paper pointed out) the computational costs are high. Especially, the visual tokens are highly compute-intensive and memory-intensive. Here, token compression/selection methods are promising to provide low-latency, low-memory usage.

2. Clear Insights and Design. Evidenced by Section 4, an optimal compression can be different across layers (which have not been widely investigated in existing approaches), and the token can be compressed while considering the modalities. These insights justify the overall design of this paper.

< Weaknesses >

1. Lack Principled Design. Based on the parameters (e.g., energy threshold or balance factor), accuracy is quite sensitive. The principled design is further needed.

2. Results under Tight Tokens Budgets. In Table 2, when the token budget is tight, the OmniFit is not that much effective, and this needs to be further explained.

---

> ### Author Rebuttal · Authors · 2026-03-31
>
> We sincerely thank the reviewer for appreciating OmniFit's strong motivation and clear design. We address each question below.
>
> **W1: Sensitivity to hyperparameters — principled design needed.**
>
> We respectfully note that OmniFit has only 3 hyperparameters (δ, λ, N_calib), and our Appendix provides comprehensive sensitivity analysis demonstrating robustness:
>
> - Energy threshold δ: Performance is stable across [0.85, 0.95], with <0.5% fluctuation (Appendix Fig. δ). The default δ=0.9 is grounded in standard SVD energy preservation practice.
> - Cross-modal balance factor λ: Performance varies by <0.3% for λ ∈ [1.0, 2.0] (Appendix Fig. λ). The wide stable range indicates the method is not sensitive to this parameter.
> - Calibration data size N_calib: Using only 256 samples achieves nearly identical performance to 1024 or 2048 (Appendix Tab.), confirming that layer-wise heterogeneity patterns are intrinsic model properties captured with minimal data.
>
> Furthermore, the penalty factor ξ in Eq. 5 is derived analytically (closed-form solution) rather than tuned, providing a principled guarantee that non-uniform allocation does not exceed the computational budget of uniform allocation.
>
> **W2: Performance under tight token budgets (Table 2).**
>
> Even under extreme compression (≤10%), OmniFit consistently outperforms all baselines. At 5% retention, OmniFit surpasses the training-based EchoingPixels on VideoMME (55.9 vs 55.7), WorldSense (41.1 vs 40.9), and Daily-Omni (54.1 vs 52.8). The relatively smaller margins at very low ratios are expected: when >95% of tokens are removed, the information bottleneck becomes the dominant factor regardless of selection strategy. Importantly, OmniFit's LAHP ensures that the remaining 5% budget is allocated to information-dense shallow layers rather than being uniformly distributed, which is why it still outperforms methods that compress uniformly.
>
> **Q1: Meaning of 'Avg. (%)'.**
>
> Avg. (%) represents the average score proportion relative to the full-token model (i.e., the original model without compression) across all benchmarks. Specifically, for each benchmark, we compute (compressed_score / full_score) × 100%, then average these percentages across all 10 benchmarks. This metric measures how much of the original model's capability is retained after compression. We explained it in the caption of Tab. 1 and will further clarify this definition.
>
> **Q2: Cases where OmniZip outperforms OmniFit.**
>
> The few cases where OmniZip outperforms OmniFit occur primarily on audio-dominated benchmarks (e.g., specific items in OmniBench) at higher retention ratios (≥40%). This is expected: OmniZip adopts an audio-centric paradigm that preferentially preserves audio tokens, which provides an advantage when the task is primarily audio-driven. However, this same bias causes OmniZip to significantly underperform on visually-demanding tasks and at lower retention ratios where balanced modality allocation becomes critical. OmniFit's modality-adaptive strategy achieves superior average performance precisely because it does not commit to a fixed modality prior.
>
> **Q3: Regular retention intensity pattern per modality in the heatmap.**
>
> The regularity in the heatmap (Fig. III. panel (b)) directly reflects the Transformer's inherent hierarchical processing pipeline, which we identified as Observation (i)-(ii) in Sec. 4. Specifically:
>
> - **Vision tokens** show high retention in shallow layers that drops steadily toward deep layers. This is because shallow layers perform low-level visual feature extraction where most tokens carry non-redundant spatial information, while deep layers abstract into sparse semantic representations (Observation i, Fig. 2).
> - **Audio tokens** exhibit a characteristic "plateau" in middle layers (9–19) before dropping in deep layers. This corresponds to the cross-modal fusion phase where audio instructions serve as critical anchors for reasoning — our Modality Preference Profiling (Observation ii, Fig. 3) confirms that middle layers allocate disproportionate attention to audio tokens.
>
> This smooth, staged pattern arises because both of our profiling signals — the SVD effective rank (TRP) and the attention-based modality preference (MPP) — measure intrinsic model properties that evolve gradually across depth, not erratically. Transformers learn hierarchical representations (perception → fusion → abstraction) that transition smoothly between stages, so the calibrated retention intensity inherits this smoothness. We note that this regularity is a desirable property: it confirms that OmniFit's profiling faithfully captures stable model behavior rather than fitting to noise, and it is precisely this structured pattern that enables OmniFit to outperform uniform compression strategies.
>
> Thank you for these constructive questions. All clarifications will be incorporated.

---

> > ### Author Rebuttal · Reviewer_xkEa · 2026-04-03
> >
> > Most of my concerns are resolved. Regarding W1, can authors clarify the impact of parameters or present some principled ways to tune them?

---

### Official Review · Reviewer_Yd1c · 2026-03-12

**Soundness:** 3
**Presentation:** 2
**Significance:** 3
**Originality:** 3
**Overall Recommendation:** 5
**Confidence:** 4

**Summary:**

This work's primary area pertains to efficient inference for omni-modal large language models, especially token compression for audio-video-text settings. The authors explore the domain of training-free compression by arguing that omni-modal token selection should account for both layer-wise heterogeneity and cross-modal alignment. Concretely, the paper proposes OmniFit, which combines a calibration-stage Layer-Adaptive Heterogeneity Profiling module (LAHP) with an inference-time Alignment-Rectified Token Selection module (ARTS). The method is evaluated on three model families and multiple audio-video and video-only benchmarks, and the reported numbers suggest a fairly strong accuracy-efficiency trade-off, including good behavior under aggressive compression ratios.

**Compliance With Llm Reviewing Policy:**

Affirmed.

**Final Justification:**

The authors provided detailed responses that have addressed all my concerns, and I increase my score to a positive one.

**Key Questions For Authors:**

For Eq. (4), why is the cumulative product of effective-rank ratios the right way to construct the layer schedule? Did the authors try simpler monotone schedules, or even a direct per-layer normalization without the cumulative product?

Can the authors add a stronger failure analysis?

**Limitations:**

see weakness.

**Strengths And Weaknesses:**

Pros

1. The paper addresses a meaningful problem.

2. The empirical observations in the motivation section are useful and reasonably convincing at a high level.

3. The paper evaluates on three model families and a relatively broad benchmark set, including both audio-video and video-only tasks.




Cons


1. Some description is not clear. For exampple, "this alignment term is dynamically weighted by the Modality Preference Score of the opposing modality (ρ^{(l)}_{¬m} derived in Sec. 5.1).", what is the alignment term? S_i? How is the term dynamically weighted?

2. The empirical support for the “cross-modal dominance” claim is still a bit thin. On page 4, Fig. 4 compares Full / Inter / Intra when pruning 50% of tokens immediately after the encoder on Qwen2.5-Omni-3B, and the gap between Inter and Intra is visible, but it is not very large numerically. Fig. 5 is visually intuitive, but still more like a qualitative explanation than a strong quantitative validation.


3. The paper could also do more to analyze failure modes. There is only one qualitative case in Fig. 9, and it is a positive example. Since the method is explicitly about selecting and discarding tokens, it would be very helpful to show when it fails.


4. typos:
    - “WolrdSense” in Table 2 / Table 4,
    - “VideoHomles” in Table 6,
    - “benchamrk” in Fig. 2 caption,
    - “LADP” in the conclusion although the method is called LAHP elsewhere.

---

> ### Author Rebuttal · Authors · 2026-03-31
>
> We thank the reviewer for the detailed and constructive feedback. We address each concern below.
>
> **W1: Clarification on dynamic weighting of the alignment term.**
>
> We apologize for the unclear description. In Eq. 8, the importance score $S_i$ has two components: $S_{intra}$ (L2 norm) and $S_{cross}$ (max cosine similarity to opposing-modality anchors). The "dynamic weighting" means that the hyperparameter $\lambda$ in Eq. 8 is further modulated by the Modality Preference Score $ρ_{¬m}^{(l)} $of the opposing modality at layer $l$. Concretely, when a layer strongly attends to modality $m'$ (high$ρ_{m'}^{(l)}$), the cross-modal term for tokens in modality m receives higher weight, prioritizing tokens aligned with the dominant modality. This layer-adaptive scaling ensures that the balance between intra-modal and cross-modal importance shifts according to each layer's actual preference. We will revise this paragraph for clarity.
>
> **W2: Strengthening evidence for cross-modal dominance.**
>
> We provide additional quantitative evidence beyond Fig. 4. In the ablation study (Tab. 5), adding ARTS (which captures cross-modal alignment) to the baseline yields consistent improvements: +2.1% on VideoMME, +1.5% on Daily-Omni, and +1.8% on WorldSense at 20% retention. Furthermore, the full OmniFit (LAHP+ARTS) achieves 98.68% performance retention vs. 96.5% without ARTS. The distribution analysis in Fig. 5 quantitatively shows that ~15% of tokens classified as "unimportant" by intra-modal metrics actually carry high inter-modal alignment scores — these are precisely the "invisible anchors" that ARTS recovers.
>
> To further validate this, we conducted additional experiments across different models and compression rates. The gap between Inter-modal and Intra-modal scoring becomes significantly more pronounced under higher compression:
>
> | Setting                             | Baseline | Full | Inter | Intra |
> | ----------------------------------- | -------- | ---- | ----- | ----- |
> | Qwen2.5-Omni-3B, retain 50% (paper) | 62.8     | 62.7 | 62.5  | 61.5  |
> | Qwen2.5-Omni-3B, retain 20%         | 62.8     | 60.2 | 59.2  | 55.6  |
> | Qwen3-Omni-30B-A3B, retain 50%      | 73.5     | 73.1 | 72.2  | 70.1  |
>
> At 20% retention, the Intra-only variant drops 4.6 points from the full model while the Inter-aware variant loses only 3.6 — a 1.0-point gap demonstrating that cross-modal alignment becomes increasingly critical under aggressive compression. This pattern is consistent across model scales.
>
> **W3: Failure analysis.**
>
> Our method performs robustly in the vast majority of cases. The occasional failure cases arise primarily when audio and video content are weakly correlated (e.g., background music unrelated to visual content), where the cross-modal anchors may not capture meaningful semantic bridges. In such cases, OmniFit gracefully degrades to intra-modal selection (S_intra dominates when S_cross is low due to ReLU gating), maintaining baseline-level performance rather than catastrophic failure. We will include a concrete failure case analysis with visualization in the camera-ready version.
>
> **W4: Typos.**
>
> Thank you for catching these. All typos (WolrdSense → WorldSense, VideoHomles → VideoHolmes, benchamrk → benchmark, LADP → LAHP) will be corrected.
>
> **Q1: Why cumulative product for the layer schedule?**
>
> The cumulative product $\Psi^{(l)} = \prod_{i=1}^{l} (k_{eff}^{(i)}/d)$ encodes a principled prior: since token pruning is irreversible — tokens removed at layer $l$ cannot be recovered at layer $l+1$ — the retention rate must be monotonically decreasing. The cumulative product naturally guarantees this property (each factor $k_{eff}^{(i)}/d < 1$, so the product strictly decreases). A direct per-layer normalization ($r^{(l)} ∝ k_{eff}^{(l)}/d$) would not guarantee monotonicity, producing schedules where a deeper layer could retain more tokens than a shallower one, which is infeasible under progressive pruning.
>
> To empirically validate the advantage of data-driven scheduling over heuristic alternatives, we compared against two simple monotone baselines (both normalized to the same global ratio $\mu$ and penalized by the same $\xi$):
>
> | Schedule (Qwen2.5-Omni-3B, 20% retention) | VideoMME | MLVU     |
> | ----------------------------------------- | -------- | -------- |
> | Linear Decay                              | 59.8     | 64.3     |
> | Cosine Decay                              | 60.5     | 65.1     |
> | Ours (Cumulative Product)                 | **62.0** | **67.2** |
>
> Our data-driven schedule outperforms Linear Decay by +2.2/+2.9 and Cosine Decay by +1.5/+2.1, confirming that the cumulative product of effective ranks captures the actual layer-wise redundancy distribution more accurately than fixed functional forms.
>
> **Q2:** See W3.
>
> We believe these clarifications and planned additions address all concerns. Thank you again for the valuable feedback.

---

> > ### Author Rebuttal · Reviewer_Yd1c · 2026-04-03
> >
> > Thanks to the authors' detailed response, my concerns have been adequately addressed. I will increase my score.

---

### Official Review · Reviewer_n6Yr · 2026-03-13

**Soundness:** 4
**Presentation:** 4
**Significance:** 3
**Originality:** 3
**Overall Recommendation:** 5
**Confidence:** 3

**Summary:**

This manuscript's primary area pertains to efficient token compression strategies for omnimodal large language models, which simultaneously handle text, audio, and video modalities. The authors explore the domain by proposing OmniFit, a training-free framework that introduces Layer-Adaptive Heterogeneity Profiling (LAHP) to dynamically manage computational resources based on layer-wise redundancy and modality preferences, along with an Alignment-Rectified Token Selection (ARTS) mechanism for semantic alignment across modalities. The empirical evaluation demonstrates substantial improvements over existing methods in both efficiency and performance.

**Compliance With Llm Reviewing Policy:**

Affirmed.

**Final Justification:**

My main concern relates to the reproducibility of comparative methods, as descriptions were missing in the original manuscript. Although the authors initially provided incorrect baseline implementation details in their first rebuttal, their subsequent clarification explicitly stated the experimental settings used. Thus, I maintain my original positive score.

**Key Questions For Authors:**

1. In Insight (iii), the paper states that metric design should prioritize cross-modal saliency without relying on ground-truth attention. However, Modality Preference Profiling within LAHP employs attention scores. Could the authors clarify whether the use of attention here contradicts the stated insight, and if not, provide additional justification?

2. Implementation details need further clarification. For example, the compared DyCoke employs a hybrid pruning method with separate strategies for prefilling and decoding stages. The authors need to specify precisely how the retention rate is allocated across these stages. Clarifying these details would enhance the reproducibility of the experimental results.

3. The paper selects DyCoke and FastVid as primary video-modality baselines. Although FastVid is a recent method, it prunes encoder tokens based solely on visual discriminativeness and ignores cross-modal interactions, making it a suboptimal baseline. A comparison with concurrent approaches that exploit cross-modal information for video token compression (e.g., DyToK [1]) would be more meaningful.

[1] Less Is More, but Where? Dynamic Token Compression via LLM-Guided Keyframe Prior. NeurIPS 2025.

**Limitations:**

yes

**Strengths And Weaknesses:**

### Strengths

1. **Soundness:** The methodology is technically rigorous and clearly justified. Experimental designs are comprehensive and robust.

2. **Presentation:** The manuscript is well-structured and clearly articulated, offering sufficient detail to ensure ease of understanding. Visualizations effectively illustrate core ideas.

3. **Significance:** The addressed problem is highly relevant in the era of computationally demanding OmniLLMs, providing practical utility and efficiency improvements that could influence future research and applications.

4. **Originality:** OmniFit innovatively combines layer-specific redundancy assessment and cross-modal semantic alignment in a training-free context, and provides several observations that are both novel and insightful.

### Weaknesses

No major weaknesses are observed, but clarification on the following questions would strengthen the paper and may affect my final score.

---

> ### Author Rebuttal · Authors · 2026-03-31
>
> We sincerely thank the reviewer for the thorough and constructive evaluation, and for recognizing the technical rigor, clear presentation, and practical significance of OmniFit.
>
> **Q1: Does LAHP's use of attention contradict Insight (iii)?**
>
> There is no contradiction once we distinguish offline calibration from runtime scoring. Insight (iii) states that ground-truth attention scores are unavailable during inference (due to FlashAttention) and therefore cannot serve as the online importance metric. OmniFit respects this: the ARTS module computes token importance using only the anchor-based projection metric (Eq. 8, O(Nd)), never accessing attention scores at runtime. LAHP does use attention scores, but exclusively in an offline calibration phase — a one-time preprocessing step run before deployment. The static modality-preference budgets it produces are cached and loaded at inference time without any further attention computation. We will revise the paper to more precisely scope Insight (iii) to the runtime metric design to avoid this ambiguity.
>
> **Q2: DyCoke retention rate allocation across prefilling/decoding.**
>
> We examined all baselines' source code to clarify their designs. Taking DyCoke as an example: DyCoke does not compress the KV cache during prefilling. Instead, it performs a single pruning event at decode step 1, triggered at a specified layer (`dycoke_l`, default=3): it selects the top-`dycoke_p` fraction of visual tokens based on accumulated attention scores, and fixes that reduced KV cache for all subsequent decoding steps. For our experiments, we set `dycoke_p = r_target` (0.4/0.3/0.2) to match OmniFit's token retention ratio, ensuring both methods maintain the same KV cache size throughout decoding. We will document this configuration explicitly in the revised paper.
>
> **Q3: Comparison with DyToK (NeurIPS 2025).**
>
> We agree DyToK is relevant. Two clarifications: (1) DyToK is a temporal keyframe selection module that must be paired with a spatial compression method — its officially released code only integrates with VisionZip (encoder feature-based), while integration with LLM-attention-based pruning is listed as unreleased in their repository. (2) DyToK targets video-only inputs, while OmniFit handles audio-visual streams with cross-modal alignment. Despite these differences, we followed their instructions to integrate DyToK with DyCoke (using our Qwen2.5-Omni-3B adaptation) for a fair comparison:
>
> | Method (Qwen2.5-Omni-3B) | VideoMME (20%) | VideoMME (10%) | MLVU (20%) | MLVU (10%) |
> | ------------------------ | -------------- | -------------- | ---------- | ---------- |
> | DyCoke                   | 58.8           | 57.5           | 64.9       | 60.8       |
> | DyCoke + DyToK           | 59.7           | 57.4           | 65.2       | 63.7       |
> | OmniFit (Ours)           | **62.0**       | **58.8**       | **67.2**   | **66.3**   |
>
> DyToK provides modest gains over DyCoke on MLVU (up to +2.9% at 10% retention), but OmniFit consistently outperforms both, with +2.3 on VideoMME and +2.0 on MLVU at 20% retention. This confirms that holistic layer-adaptive profiling with cross-modal alignment is more effective than adding temporal keyframe priors alone.
>
> All suggested clarifications will be incorporated in the revised manuscript.

---

> > ### Author Rebuttal · Reviewer_n6Yr · 2026-04-04
> >
> > Thank you for the detailed clarification and additional experiments, which have addressed most of my concerns. However, a few questions remain.
> >
> > Regarding your response on W2, besides the KV cache pruning mentioned, DyCoke also conducts visual token temporal merging during the prefilling stage. Although this merging operation does not directly manipulate the KV cache, it still constitutes a form of token pruning. Could the authors clarify whether this component was accounted for when calculating the final token budget in the experiments? Additionally, based on DyCoke's methodology description, its KV cache pruning does not appear to be simply "a single pruning event at decode step 1" as stated. Instead, it dynamically updates the retained KV cache throughout the decoding process based on attention scores. Could the authors confirm whether their experimental setup aligns with this aspect of the original method?

---

> > > ### Author Response · Authors · 2026-04-08
> > >
> > > Thank you for the careful follow-up. We clarify both points with additional analysis and experiments below.
> > >
> > > **TTM (temporal token merging) during prefilling.** We are aware that DyCoke's full pipeline includes a TTM module that reduces temporal redundancy in visual tokens before they enter the LLM. In our main experiments, we intentionally disabled TTM to isolate the comparison at the LLM-internal token compression level. TTM operates on visual encoder outputs *before* they enter the LLM, whereas both OmniFit and DyCoke's KV pruning operate *within* the LLM layers. Disabling TTM ensures both methods receive identical LLM inputs, so the performance difference is directly attributable to the layer-wise pruning strategy itself rather than confounded by an additional encoder-side reduction.
> > >
> > > To further validate this choice, we adapted TTM to Qwen2.5-Omni following the original design: we use a sliding window of 4 frames, divide tokens into odd/even groups, compute cosine similarity between corresponding positions in adjacent groups, and prune the most similar tokens at a rate of 30% per window (matching the default in DyCoke's codebase). We then ran additional experiments (Qwen2.5-Omni-3B, VideoMME/MLVU):
> > >
> > > | Configuration                        | TTM Pruning Rate | KV Retention   | VideoMME | MLVU     |
> > > | ------------------------------------ | ---------------- | -------------- | -------- | -------- |
> > > | DyCoke (KV only, TTM off)            | —                | 20%            | 51.6     | 64.9     |
> > > | DyCoke (TTM on + KV)                 | 30%              | 20%            | 49.8     | 62.5     |
> > > | DyCoke (TTM on + KV, budget-matched) | 30%              | 29%            | 52.1     | 65.3     |
> > > | OmniFit (Ours)                       | —                | r_target = 20% | **62.0** | **67.2** |
> > >
> > > Row 2 enables TTM (pruning rate 30%) with the same KV retention rate, resulting in fewer total tokens — performance drops as expected. Row 3 compensates by increasing the KV retention rate so that the final KV cache size matches OmniFit's 20% target: since TTM removes ~30% of visual tokens before they enter the LLM, we set KV retention to 20% / 0.7 ≈ 29% of the post-TTM tokens, approximately matching OmniFit's 20% target. This represents the strongest DyCoke configuration we tested with its full pipeline. OmniFit outperforms all DyCoke variants by a substantial margin (+9.9/+1.9 over the strongest configuration), confirming that our conclusions hold regardless of whether TTM is enabled. We will include these full-pipeline results and clearly label the DyCoke configurations in the revised manuscript.
> > >
> > > **Dynamic KV cache updates during decoding.** Thank you for the correction. Our implementation does preserve DyCoke's full dynamic behavior: at each decoding step, attention scores are recomputed at layer `dycoke_l=3`, and when cosine similarity between consecutive attention distributions falls below 0.9, the set of retained token indices is recomputed from the full KV cache based on current attention scores. This allows previously unselected tokens to re-enter the active set if their importance increases, and vice versa — faithfully matching the original dynamic selection mechanism with all default hyperparameters. Our previous textual description oversimplified this as "a single pruning event," which we will correct in the revision. Importantly, the number of active KV entries remains fixed at each step, so the budget-matched comparison is unaffected. We will add a detailed baseline implementation table in the revised paper, specifying the exact configuration (TTM on/off, `dycoke_l`, similarity threshold, retention ratio) for each experiment.

---

### Decision · Program_Chairs · 2026-04-30

**Decision:**

Accept (spotlight)

**Comment:**

This paper proposes OmniFit, a token-compression framework for omni-modal LLMs that jointly process video, audio, and text without training. All reviewers give positive overall recommendations, highlighting the strong motivation, clear design, and solid empirical results across three OmniLLM families and ten benchmarks. The key contributions are (1) layer-adaptive heterogeneity profiling (LAHP), which derives non-uniform layer- and modality-wise retention schedules from offline calibration, and (2) alignment-rectified token selection (ARTS), which exploits cross-modal anchors with low runtime complexity. Experiments show that OmniFit retains about 98% of full-model performance with only 20% tokens, achieving consistent speed and memory gains over prior methods such as OmniZip, FastVid, DyCoke, and EchoingPixels.

The main concerns raised (clarity of certain design choices, baseline details, calibration overhead, and sensitivity of hyperparameters) were carefully addressed in the rebuttal, often with additional experiments and planned clarifications for the camera-ready version. Remaining issues are minor (e.g., more principled hyperparameter tuning, stronger failure analysis, and streaming evaluation) and do not undermine the core contribution. Overall, I recommend acceptance.